# Should You Use Your Large Language Model to Explore or Exploit?

## Abstract

We evaluate the ability of the current generation of large language models (LLMs) to help a decision-making agent facing an exploration-exploitation tradeoff. We use LLMs to explore and exploit *in silos* in various (contextual) bandit tasks. We find that while the current LLMs often struggle to exploit, in-context mitigations may be used to substantially improve performance for small-scale tasks. However even then, LLMs perform worse than a simple linear regression. On the other hand, we find that LLMs do help at exploring large action spaces with inherent semantics, by suggesting suitable candidates to explore.

## 1 Introduction

There's been significant interest in the machine learning community to apply recent advances in generative AI and large language models (LLMs) to solve important decision-making problems. Early work in this direction has already produced impressive agentic behavior in both virtual (e.g., Wang et al., 2023; OpenAI, 2025) and physical-world environments (e.g., Black et al., 2024).

Beyond generalization (needed for supervised learning), decision-making under uncertainty requires two additional capabilities: *exploitation* (making the best decision given the current data) and *exploration* (trying new options for long-term benefit). Balancing the two has led to an enormous literature (e.g., see books Slivkins, 2019; Lattimore & Szepesvári, 2020; Agarwal et al., 2020).

A recent line of work (e.g., Krishnamurthy et al., 2024; Nie et al., 2024) evaluates the ability of LLMs to balance exploration and exploitation entirely *in-context*, i.e., specifying the problem description, parameters, and history in the LLM prompt. Focused on simple tasks in reinforcement learning (RL), these results are mixed. Both papers show that LLMs fail to solve these tasks adequately out-of-the-box, but they can be prompted to do so by providing succinct summary statistics in-context. However, such statistics do not exist beyond simple decision-making problems, e.g., for contextual bandits. RL-specific pre-training or fine-tuning (on data from algorithmic baselines on similar problem instances) tends to work well (see Related Work), but may be prohibitive due to cost or insufficient training data. Besides, using a commonly available frontier LLM would leverage its "generalist" intelligence and may be much easier in terms of logistics and required expertise.

Motivated by these observations, we study the ability of LLMs to explore and exploit in-context *in silos*, with an eye towards leveraging a pre-trained LLM (and the inductive bias therein) as a part of a larger decision-making agent. We focus on (contextual) bandits, as a standard abstraction for the explore-exploit tradeoff. We experiment with a variety of LLMs: GPT-4, GPT-4O, GPT-3.5, QWEN-2.5, GEMMA-3, MISTRAL-7B, as well as DEEPSEEK-R1-DISTILL-QWEN, a reasoning model. [1]

In Section 2, we evaluate LLMs as *exploitation oracles* for contextual bandits. Given a history of (context, action, reward) tuples, the LLM is tasked with identifying the best action to take given a new context. Our results here are mixed. We show that LLMs can effectively exploit in-context for

---

[1] Several other LLMs did not work for our purposes: LLAMA-3.2-3B and LLAMA-3-8B would not follow instructions, as well as DEEPSEEK-R1-DISTILL-LLAMA-70B (a reasoning model); QWEN-2.5-MATH-70B (another reasoning model that we tried) has a short context window. We restrict our GPT access to GPT-4, GPT-4O, and GPT-3.5, as newer variants such as GPT-4.1 and GPT-4.5 are primarily optimized for efficiency (e.g. latency, cost, coding) rather than broad general reasoning, and GPT-5 uses a reasoning backbone as a routing mechanism across sub-models.

small-sized problems, but their performance degrades when the problem becomes moderately sized. We find that in-context summary techniques are useful for improving performance, but LLMs with these mitigations still perform worse than a simple linear regression baseline.

In Section 3, we evaluate LLMs as *exploration oracles* which suggest candidate actions by discretizing a large action space. To do so, we introduce a text-based multi-armed bandit task, where actions correspond to free-text answers to an open-ended question, and rewards are driven by the distance from some preselected answer in an embedding space (the latter is computed exogenously). Given the high dimensionality of the action space, traditional discretization approaches typically used in continuous bandit problems are inapplicable. However, an LLM can generate a small set of candidate actions which can then be used to instantiate an off-the-shelf bandit algorithm. We experiment with several prompting strategies, and find that they all lead to relatively good exploration. Finally, we repeat our experiments on a larger-scale bandit task based on paper titles and abstracts from arXiv (where the goal is to find a suitable title for a given abstract), with similar findings.

**Related work.** Our results belong to a growing line of work on using pre-trained LLMs for in-context reinforcement learning (RL). Coda-Forno et al. (2023); Krishnamurthy et al. (2024); Nie et al. (2024); Monea et al. (2024); Xia et al. (2024); Park et al. (2024); Wu et al. (2024) evaluate the ability of LLMs to solve various multi-armed bandit and contextual bandit tasks, and find that the current generation of LLMs largely fail to solve these tasks in-context. Indeed, positive findings are restricted to very simple tasks and/or require substantial mitigations (which in turn do not readily extend beyond simple settings). Xia et al. (2024) use LLMs to solve dueling bandit tasks, and Park et al. (2024) also evaluate the ability of LLMs to learn in games. While our paper is primarily concerned with whether LLMs succeed as algorithms, several others (e.g., Schubert et al., 2024; Hayes et al., 2024; Coda-Forno et al., 2024) use in-context bandits (and many other tasks) to study whether LLMs exhibit human-like behavior/biases in decision-making.

A broader literature on in-context learning (starting from Brown et al., 2020) aims to solve various tasks by providing all relevant information in the LLM prompt. The work on *exemplar selection* (selecting examples and other information to present in-context) (e.g., Khalifa et al., 2023; Zhang et al., 2022; Xiong et al., 2023; Tonglet et al., 2023) is relevant to our exploitation experiments.

A growing line of work aims to use LLMs as a part of a larger decision-making agent (e.g., Li et al., 2024; Zhou et al., 2023; Zhao et al., 2024). Our exploration experiments take inspiration from the work on using LLMs as "action priors" inside of a larger RL algorithm (Yan et al., 2024; Carta et al., 2023; Yao et al., 2024; Hao et al., 2023). Much of this work falls under the proposer-verifier framework of Snell et al. (2024), where an LLM proposes several possible sequences from which a verifier selects suitable candidates. In comparison, our goal is a more systematic evaluation of LLMs' abilities to explore large action spaces, in isolation from other components of the decision-making task.

Finally, a parallel line of work trains transformers to solve various RL tasks (e.g., Laskin et al., 2022; Lin et al., 2023; Lee et al., 2024; Raparthy et al., 2023; Xu et al., 2022; Lehnert et al., 2024; Mukherjee et al., 2024). Our negative results in Section 2 provide an additional foundation for this work, emphasizing the shortcomings on frontier LLMs that are not custom-trained for RL.

**Background on multi-armed bandits (MAB).** We consider tasks based on MAB and contextual bandits (CB), well-studied special cases of RL that abstract the explore-exploit tradeoff, see Slivkins (2019); Lattimore & Szepesvári (2020) for background. In MAB, there are $T$ rounds and $K$ arms. In each round $t \in [T]$, the learner chooses an action (*arm*) $a_t \in [K]$ and observes reward $r_t$ drawn from some sub-Gaussian reward distribution for this arm. The reward distribution, and particularly its mean $\mu(a_t)$, are unknown to the algorithm. In CB, the learner additionally observes a context $z_t$ before each round $t$, and the expected reward $\mu(z_t, a_t)$ depends on both the context and the arm. The learner's goal is to balance exploration and exploitation to maximize cumulative reward.

An "exploitation oracle" (which optimizes for the current round given the history) naturally plugs into standard bandit algorithms such as Epsilon-Greedy, Explore-then-Commit, and Follow-The-Perturbed-Leader. Typical implementations in CB involve model-based (e.g., linear) regression or cost-sensitive classification (Slivkins, 2019, Ch.8). Designing CB exploitation oracles for their own sake, a.k.a. *offline policy optimization*, is well-studied (starting from, e.g., Beygelzimer & Langford, 2009; Dudík et al., 2014). Usage of an LLM as an "exploration oracle" is closely related to the literature on dynamic pricing and Lipschitz bandits (e.g., Kleinberg & Leighton, 2003; Kleinberg et al., 2008; 2019; Bubeck et al., 2011), as we elaborate in Section 3.

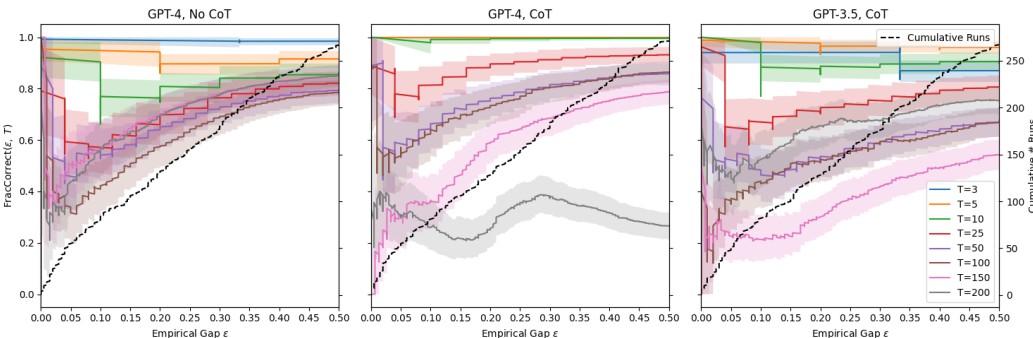

Figure 1: MAB exploit puzzle for GPT-4 (left), GPT-4 with CoT (middle), and GPT-3.5 with CoT (right), all with "buttons" prompt. The following conventions apply to all figures in this section. Each line corresponds to a particular value of #rounds $T$ and plots $\text{FracCorrect}(\epsilon, T)$ against empirical gap $\epsilon$ on the X-axis. The shaded band around the line represents a $95\%$ confidence interval. The dashed line is the number of tasks ("runs") with empirical gap $\leq \epsilon$; the resp. Y-scale is on the right.

## 2 LLMS AS EXPLOITATION ORACLES

We evaluate the ability of LLMs to *exploit* in decision-making tasks with statistical uncertainty on the outcomes. We present LLMs with in-context exploit tasks inspired by multi-armed bandits (MAB) and contextual bandits (CB). In a CB exploit task, an LLM is given a history consisting of context-arm-reward tuples, and is instructed to take the best arm given the current history and the current context. A MAB exploit task is the same, but without contexts. These tasks are generated from some parameterized distributions called *exploit puzzles*.

As our results here are largely negative, we focus on relatively small tasks. We use synthetic data, to better control task size/difficulty and trace the "boundary of tractability". We find that LLMs are consistently outperformed by a simple linear regression baseline, even on moderately-sized tasks. Due to computational constraints, our experiments focused on GPT-4, GPT-4O, and GPT-3.5, with additional LLMs evaluated as a robustness check in Appendix A.4, with similar findings.

**MAB exploit puzzles.** Our MAB-based experiments on GPT-4 and GPT-3.5 provide a partial explanation for why the current LLMs fail to solve MAB tasks in-context when presented with raw (non-summarized) history, as first observed by Krishnamurthy et al. (2024); Nie et al. (2024). Following these two papers, we try two prompts: one in which arms correspond to pushing different colored buttons and one where they correspond to showing different advertisements to users. The LLM is asked to choose the arm with the highest empirical reward in the next round. We also try chain-of-thought (CoT) prompts, for the total of 4 prompt designs: { buttons, adverts } × { CoT, no-CoT }. See Appendix A for more details on our experimental setup.

We consider an MAB exploit puzzle parametrized by gap $\Delta \in [0, 1]$ and history size $T$. The tasks, all with 5 arms, are constructed as follows. We pick an arm $a^*$ uniformly-at-random (u.a.r.). Expected rewards are assigned as $\mu(a^*) = 1/2 + \Delta/2$ and $\mu(a^*) = 1/2 - \Delta/2$ for all other arms $a$. Then, we generate a history of $T$ rounds for each arm $a$, where the reward $r_t(a)$ at each round $t \in [T]$ is an independent Bernoulli draw with mean $\mu(a)$. For a given $T$, we generate 10 tasks from this puzzle for each value of $\Delta \in \{0, .05, .1, .15, .2, .25, .3, .4, .45, .5\}$.

Given a (realized) exploit task, we measure its difficulty via *empirical gap* $\epsilon$: the difference between the largest and second-largest average reward $\bar{r}_t(a) := \frac{1}{T} \sum_{t \in [T]} r_t(a)$ among all arms $a$. Intuitively, puzzle difficulty decreases with $\epsilon$ (as one needs to distinguish the top two arms). Empirical gap mirrors the "gap" between the top two expected rewards, a standard notion of difficulty in MAB. Let $S(\epsilon, T)$ be the set of all tasks with empirical gap at most $\epsilon$ and history size $T$.

We measure an LLM's performance over a given set $S$ of tasks as the fraction of tasks for which the LLM returns a "correct answer": an arm with the largest empirical reward; denoted by $\text{FracCorrect}(S)$. We are interested in how $\text{FracCorrect}$ varies depending on the difficulty level. Hence, we plot $\text{FracCorrect}(\epsilon, T) := \text{FracCorrect}(S(\epsilon, T))$ against empirical gap $\epsilon$.

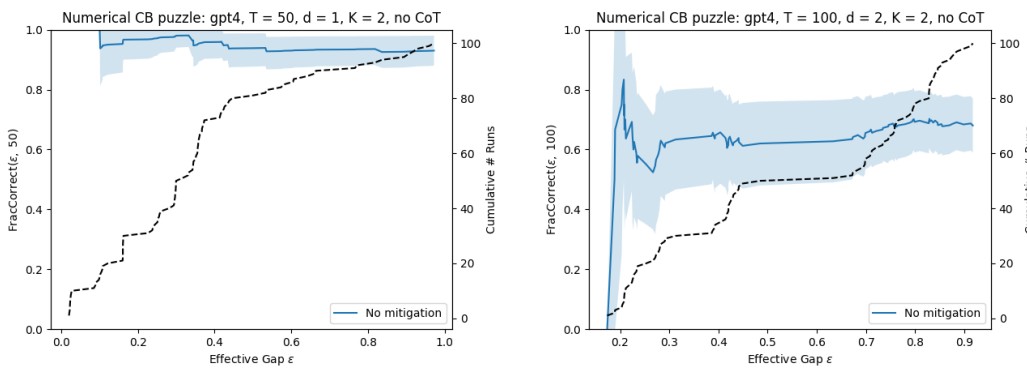

Figure 2: GPT-4 succeeds on a small CB exploit puzzle (left), but fails on a slightly larger one (right).

We find that GPT-4 and GPT-3.5 do not perform well on these MAB exploit puzzles, see Figure 1. Performance tends to degrade (1) as the history length $T$ increases and (2) as the empirical gap decreases. (Our intuition for the former is that the LLM has more data points to process in context, and for the latter is that the bandit instance becomes more difficult.) While GPT-4 generally performs much better than GPT-3.5, we found that prompting the LLM to use chain-of-thought (CoT) reasoning provided a slight boost for GPT-3.5, while hurting the performance of GPT-4.[2] We found that performance was similar across all of our prompt designs (see Appendix A for more plots).

**CB exploit puzzles and mitigations.** While the history in $K$-armed bandits can be summarized using $2K$ numbers (for each arm, the average reward and #plays) such succinct summary statistics may not be readily available (or even exist) in more complicated decision-making tasks such as CB.[3]

We focus on linear CB, where the expected reward of each arm $a$ is linear in the context $z \in \mathbb{R}^d$: $\mu(z, a) = \langle z, \theta_a^* \rangle$ for some fixed (but unknown) parameters $\theta_a^* \in \mathbb{R}^d$. [4] (We try non-linear CB, too.)

We consider a CB exploit puzzle parameterized by #arms $K$, dimension $d$, and history size $T$. The tasks are constructed as follows. We sample parameters $\theta_a \in [-1, 1]^d$ and $\gamma_a \in [-0.25, 0.25]$ independently and u.a.r. for each arm $a$. Given context $z \in \mathbb{R}^d$, expected reward for arm $a$ is $\mu(z, a) = \langle z, \theta_a \rangle + \gamma_a$. We generate a history of $T$ rounds $t \in [T]$. Contexts $z_t$ are sampled independently and u.a.r. from $[-1, 1]^d$. For simplicity, the history contains rewards of all arms $a$ in each round $t$, where the reward equals $\mu(z_t, a)$ plus an independent unit-variance Gaussian. Given the history and a new context $z_{T+1}$ (drawn in the same way), the LLM is asked to select the action for round $T + 1$ which appears best. This gives one exploit task. We generate $N$ tasks for the same $K, T, d$.

We use a modified "buttons" prompt, where contexts correspond to "numbers on a screen" which affect the payoffs for each button. The prompt does not mention linearity of the CB instance (because such model-based information is typically unavailable in applications).

Given an exploit task, we now define a "correct answer" as an arm $a$ which maximizes expected reward $\mu(z_{T+1}, a)$. [5] FracCorrect($S$) is the fraction of correct answers in a given set $S$ of tasks. Likewise, the task difficulty is not easily defined in terms of the realized rewards. Instead, we focus on the *effective gap*: the difference in expected reward between the best and second-best arm given the current context $z = z_{T+1}$. That is, the difference between the largest and second-largest number among $\mu(a, z), a \in [K]$. Intuitively, smaller effective gap corresponds to increased difficulty.

We are interested in how FracCorrect varies with effective gap. In each plot, we fix the number of arms ($K$) and dimension ($d$), and let $S(\epsilon, T)$ be the set of all tasks with given $K, d, T$ and effective gap at most $\epsilon$. We plot FracCorrect($\epsilon, T$) := FracCorrect($S(\epsilon, T)$) against $\epsilon$.

---

[2] From CoT logs, we found that GPT-4 was doing incorrect reasoning in a non-trivial fraction of runs.

[3] This consideration also motivates MAB exploit puzzles with raw (non-summarized) history, as a simpler special case of the general scenario when succinct summarization is unavailable.

[4] Linear CB are well-studied, starting from Li et al. (2010); Chu et al. (2011); Abbasi-Yadkori et al. (2011).

[5] Note that it is unclear how to define an "empirically best arm" given a CB history and the current context.

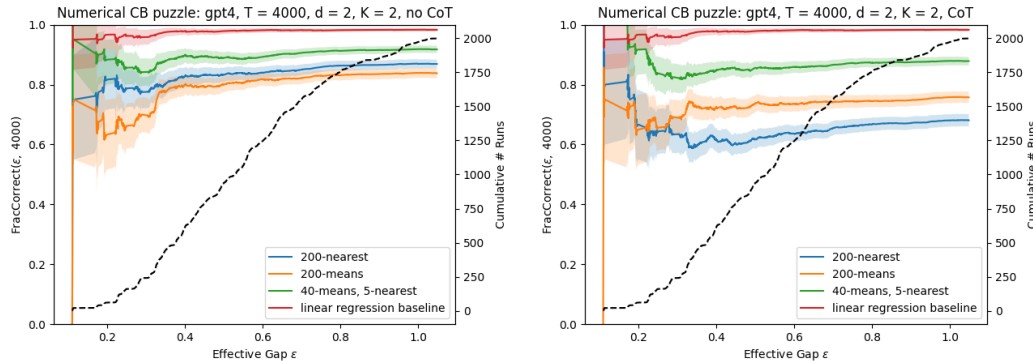

Figure 3: CB exploit puzzle with $d = K = 2$ and $T = 4000$: **mitigations help substantially**. GPT-4 without CoT (left) and GPT-4 with CoT (right). Note that providing the full history with this $T$ vastly exceeds the context window for GPT-4, GPT-4O, and GPT-3.5.

First, we find that GPT-4 obtains near-perfect performance on a "small" puzzle with $K = 2$ arms, context dimension $d = 1$, and history size $T = 50$ (Section 2). However, its performance degrades as the problem size increases: see Section 2 for $K = d = 2$ and $T = 100$. Moreover, limited prompt size may prevent processing larger histories.[6]

Motivated by these observations, we implement several natural *mitigations* inspired by the literature on exemplar selection for in-context learning (discussed in Related Work).

**1.** *$k$-nearest:* Among the observed contexts, consider the distinct $k$ contexts closest to $z_{T+1}$, according to the $\ell_2$ metric. Limit the history reported in the prompt to (the rounds with) these $k$ contexts.

**2.** *$k$-means:* Run an off-the-shelf algorithm for $k$-means clustering on contexts $\{z_1, \ldots, z_T\}$, obtaining $k$ centroid contexts $z_i^*$ and their respective clusters $Z_i^*$, $i \in [k]$. For each centroid $z_i^*$ and each arm $a$, let $\bar{r}(z_i^*, a)$ be the average reward for this arm over all rounds $t$ with contexts $z_t \in Z_i^*$. Report $(z_i^*, a, \bar{r}(z_i^*, a))$ as a context-arm-reward triple.

**3.** *$k$-means, $k' < k$-nearest:* First, run the $k$-means mitigation. Report $(z^*, a, \bar{r}(z^*, a))$ as a context-arm-reward triple, for each arm $a$ and each centeroid context $z^*$ among the $k'$ centroids closest to $z_{T+1}$ (according to the $\ell_2$ metric).

We do not explain the "mitigation strategy" in the prompt: we present the reported context-arm-reward tuples as if it were the entire history, without mentioning clustering or averaging.

Figure 3 visualizes the performance of these mitigations on a slightly larger (but still relatively small) puzzle with $K = d = 2$ and $T = 4000$. We use GPT-4 with and without CoT. We compare the mitigations against linear regression baseline (which is effectively an "upper bound", as the underlying CB instance is linear). Without CoT prompting, we find that all three mitigations achieve FracCorrect around $80\% - 90\%$, although this dips to around $60\% - 85\%$ when using CoT. In addition to (potentially) improving performance, mitigations can also offer a practical way to solve decision-making tasks using LLMs when the history is large; when $T = 4,000$, our prompt vastly exceeds the context window of all models we had access to.

However, current LLMs struggle to exploit on even moderately-sized problems, even with these mitigations. In Figure 4, we plot the performance of GPT-4O with mitigations under various hyperparameters. While $k$-means (left) is almost as bad as random guessing, the $k$-nearest mitigations (center) achieve about $50\%$ FracCorrect, and the $k$-means, $k'$-nearest mitigations (right) obtain approximately $60\%$ FracCorrect. Both our $k$-nearest and $k$-means, $k'$-nearest mitigations significantly out-perform unmitigated GPT-4O, but fall significantly short of the linear baseline.

**Reasoning models** are LLMs specifically trained to follow multi-step chains of thought to solve complex problems that require intermediate reasoning. We ran one such model, DeepSeek-R1-Distill-Qwen-32B (henceforth DEEPSEEK-R1-DISTILL-QWEN), a 32B parameter reasoning model

---

[6]E.g., our LLM access points bottomed out at $T \approx 100$-$200$ for GPT-4 and $T \approx 1000$-$2000$ for GPT-4O.

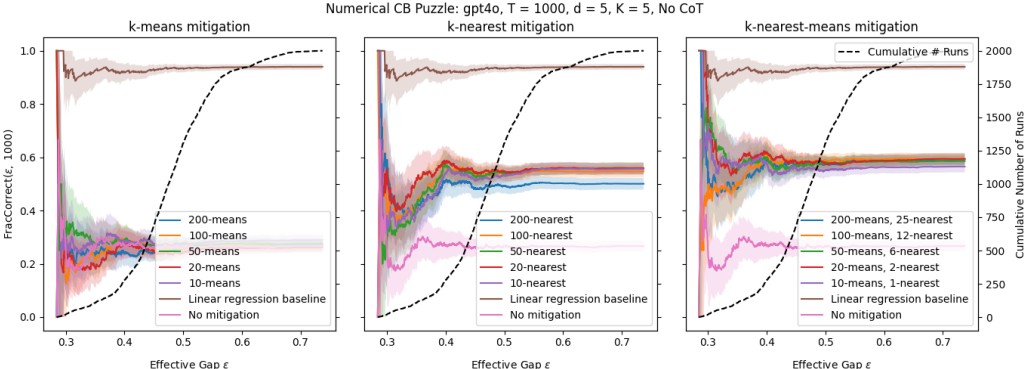

Figure 4: CB exploit puzzle with $d = K = 5$ and $T = 1000$: **mitigations perform badly**, but (mostly) much better than the no-mitigation baseline. GPT-4O without CoT.

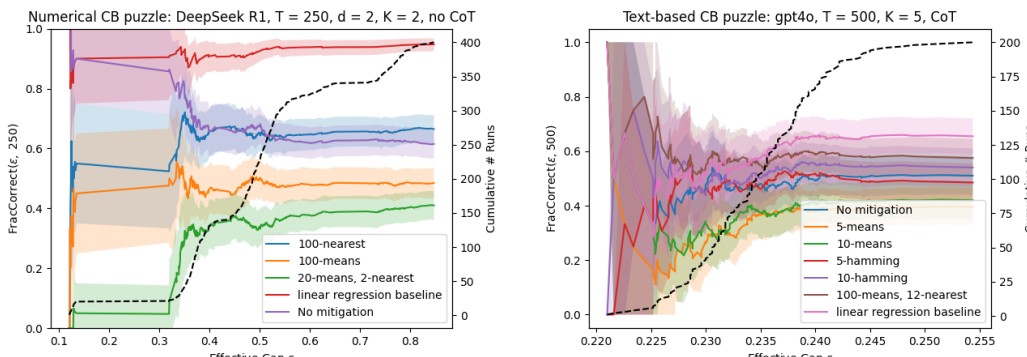

Figure 5: Left: Performance of Deepseek-R1 on our numerical CB puzzle. Right: GPT-4O on the text-based CB exploit puzzle. Some mitigations help, but are **outperformed by linear regression**.

distilled from QWEN-2.5. Our results are shown in Figure 5 (left). As before, we find that DEEPSEEK-R1-DISTILL-QWEN fails to match the performance of the baseline, even when using the mitigations. High inference costs and small context windows prevented us from running larger reasoning models.

**CB exploit puzzles (text-based & non-linear).** As a robustness check, we repeat our CB experiments on a text-based exploit puzzle. In this puzzle, contexts are items in a room (e.g. animals, objects on a table), and actions have an associated semantic meaning (e.g. eat the food item, leave the room). Rewards are still presented numerically, and are non-linear functions of both the context and action. See Appendix A for full details on our experimental setup.

Figure 5 (right) shows the performance of GPT-4O (with mitigations) on this puzzle. While the reward function is non-linear (and thus the linear baseline only achieves $70\%$ FracCorrect), we find that all configurations are still significantly out-performed by the linear baseline.

Our intuition for these findings is as follows: While LLMs are generally good with textual inputs, the reward data is numerical, so the LLM faces most/all of the same challenges as before (i.e., choosing a "solution strategy" and executing it correctly), exacerbated by non-linearity. Meanwhile, linear regression is known to often work fairly well even on non-linear data.

**Takeaways.** While unmitigated performance drops as $d, K, T$ grow, increases in $T$ can largely be taken care of by our mitigations, whereas increases in $d$ and $K$ cannot. Indeed, taken together, our findings in this section suggest that while the current generation of LLMs are effective at exploiting in small-scale decision-making tasks, they struggle in larger settings.

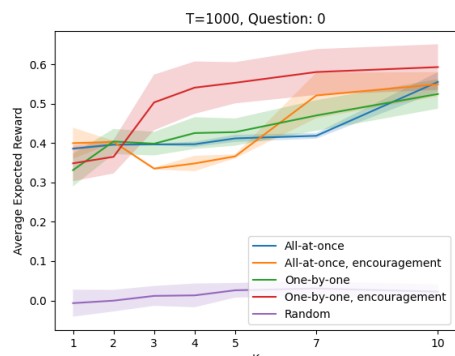 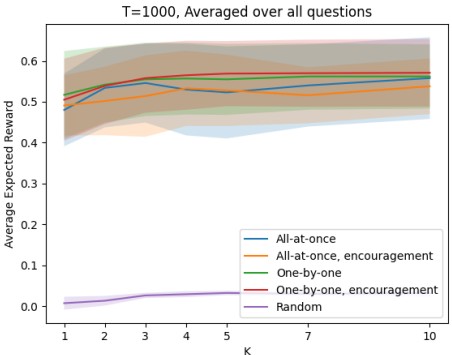

Figure 6: Algorithm's average expected reward $\overline{\texttt{rew}}(\text{task}, K)$ (averaged over rounds and over runs), against $K$, the number of candidates. Each line corresponds to a prompting strategy or the `Random` baseline. The shaded regions represent a $95\%$ confidence interval.

## 3 LLMs AS EXPLORATION ORACLES

We now study the ability of LLMs to explore large action spaces. We leverage the inductive bias of an LLM to generate a small set of candidate actions from a text-based action space, before running an off-the-shelf MAB algorithm on this set. We refer to this LLM usage as *exploration oracle*.

Our usage of LLMs for exploration is closely aligned with the idea of "discretization" in the study of dynamic pricing and Lipschitz bandits (e.g., Kleinberg & Leighton, 2003; Kleinberg et al., 2008; 2019; Bubeck et al., 2011; Slivkins, 2014). Faced with a very large action space, it may be advantageous to focus on a much smaller set of candidate actions. A naively constructed discretization, e.g., uniform with respect to some known notion of distance in the action space, may have a poor trade-off between size (#candidates) and quality (of the best candidate), placing all but a few candidates in low-performing regions of the action space. A crucial theme in this literature is "smarter" discretization algorithms which gradually "zoom in" on better-performing regions of the action space. The intuition is that if the better-performing arms comprise a low-dimensional region of a high-dimensional action space, covering this region should require much fewer samples for the same "discretization quality". Our hope is that LLMs can suggest candidate actions from this region, if there is sufficient semantic meaning encoded in the bandit task.

We consider two types of exploration tasks: answering an open-ended "philosophical" question (Q/A task) and suggesting a title for an arXiv research paper based on its abstract (arXiv task). Particular workloads within each task type are called *explore puzzles*.

The Q/A task (resp. arXiv task) is constructed as follows. We define the "best arm" $a^*$ as a contrarian answer generated by another LLM (resp. the actual title of the research paper). The expected reward $\mu(a)$ of an arm $a$ is the cosine similarity between $a$ and $a^*$ in the embedding space.[7] Here, we generate sentence embeddings using the Sentence-BERT embedding model Reimers (2019).[8] The realized reward in a given round is an independent Bernoulli sample with the mean defined above.

Since action spaces are extremely large for these tasks, standard bandit algorithms (e.g., Thompson Sampling, UCB1, etc.) catastrophically fail, and so do bandit algorithms which randomly subsample the action space. If the embedding space, the distance notion therein, and the reward-distance relation were known to the algorithm/agent (assumptions we do not make), one could, in principle, apply the "smarter" discretization algorithms mentioned above. However, the regret of these algorithms scales exponentially in the dimension $d$ of the metric space making them impractical in high-dimensional action spaces such as ours. (Indeed, our embedding space has $d = 384$.)

Like for our exploit puzzles, we highlight our results for GPT models in the main body and include the rest in Appendix B.5. Our findings are largely consistent across different LLMs.

---

[7]While cosine similarity ranges on $[-1, 1]$, it was usually strictly positive in our experiments. In the (very rare) cases where it was negative we defined the expected reward as zero.

[8]Our results are largely unchanged under the Universal Sentence Encoder Cer (2018), see Appendix B.

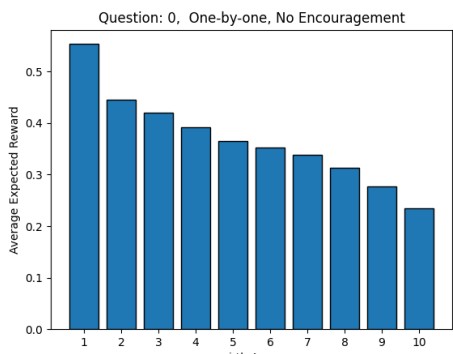 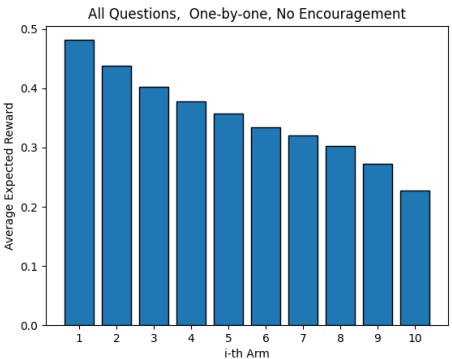

Figure 7: Arm histogram for one particular question ("What is the meaning of freedom?", left) and averaged over all questions (right). We consider $K = 10$ suggestions generated by our one-by-one prompt without encouragement. For a given "run", the $K$ suggestions are ranked by the expected reward, and then the $i$-th expected reward is averaged over all runs, for each $i \in [K]$.

**Explore puzzle: open-ended questions.** We used GPT-4 to generate a dataset of 10 open-ended questions with many reasonable answers, along with an intentionally contrarian answer for each question to serve as the ground truth. (E.g., "What does it mean to live a fulfilling life?" "Fulfillment comes from embracing discomfort.") Each question-answer pair yields a task, as defined above.

We evaluate GPT-4O, QWEN-2.5, GEMMA-3, and MISTRAL-7B as exploration oracles. We prompt them to suggest $K \in \{1, 2, 3, 4, 5, 7, 10\}$ candidate answers given a question. To grade the entire candidate set (not just the best answer), we run an off-the-shelf MAB algorithm for some large-but-realistic time horizon $T$ over these $K$ candidates. (We use the UCB1 algorithm (Auer et al., 2002) and $T = 1000$.) We record the expected time-averaged reward $\texttt{rew} := \frac{1}{T} \sum_{t \in [T]} \mu(a_t)$, where $a_t$ is the arm selected in round $t$. We repeat this process (candidate selection and running UCB1) 10 times for a given task and $K$, and record the average $\texttt{rew}$ over these runs, denoted by $\overline{\texttt{rew}} = \overline{\texttt{rew}}(\text{task}, K)$.

We try several prompting strategies. We either ask the LLM to generate suggestions "all-at-once" with temperature 0 or "one-by-one" with temperature 1 (we repeatedly show the LLM the list of candidate answers so far and ask it to generate one more). We also experiment with explicitly prompting the LLM to provide a diverse set of candidate answers ("with encouragement"). Thus, we have 4 prompting strategies: { all-at-once, one-by-one } × { with, without } encouragement. To compare against *not* using an LLM as an exploration oracle, we also consider a baseline (`Random`) in which the candidates are $K$ points selected independently and u.a.r. in the embedding space.

We visualize our findings in Figure 6. We plot $\overline{\texttt{rew}}(\text{task}, K)$ against $K$, for one particular task (left) and averaged across all tasks (right).[9] Each line corresponds to a particular prompting strategy or the `Random` baseline. We find that all four prompting strategies perform similarly, with average reward between 0.5 and 0.6, and typically peak in performance around $K = 3$ or $K = 4$ suggestions. In contrast, the `Random` baseline catastrophically fails, with its average reward never exceeding 0.1. We conclude that the LLM does succeed as an exploration oracle.

We observe that the LLM-generated suggestions pass the "eye test", in the sense that we get reasonable, yet spiritually and semantically different answers for a given question. For example, given the question "What is the role of technology in society", the first $K = 5$ suggestions generated by our one-by-one prompt are as follows: (1) Facilitates communication, innovation, and efficiency, (2) Transforms daily life and shapes culture, (3) Drives connectivity and enhances productivity, (4) Facilitates control and surveillance, (5) Disrupts traditional relationships and norms.

Finally, we verify that the candidate suggestions are substantially *different* from one another. Figure 7 visualizes the spread of expected rewards within the candidate set. We consider $K = 10$ suggestions generated by our one-by-one prompt without encouragement. For a given "run" of the candidate selection, the suggestions are ranked by the expected reward, and then the $i$-th expected reward is averaged over all runs, for each $i \in [K]$.

---

[9] See Appendix B for similar plots for the 9 other tasks.

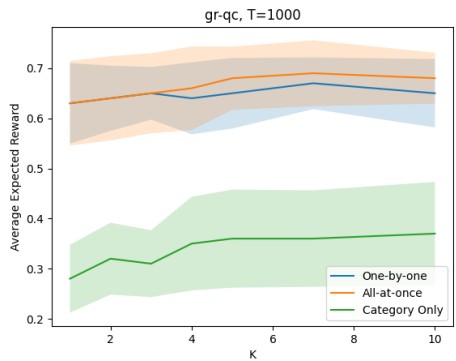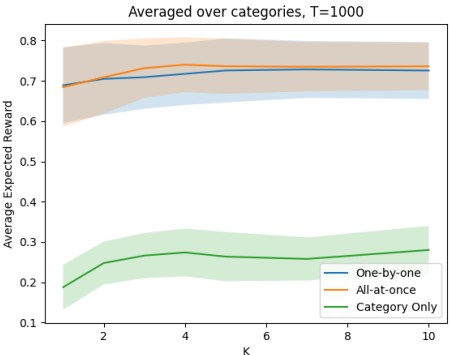

Figure 8: Algorithm's average expected reward $\overline{\mathtt{rew}}$(category, $K$) (averaged over rounds and tasks), against $K$, the number of candidates. Each line corresponds to a prompting strategy or the Category-Only baseline. The shaded regions represent a $95\%$ confidence interval. A single arXiv category ("General Relativity and Quantum Cosmology", left), averages over 6 categories (right).

A more detailed comparison between the prompting strategies is not fruitful for these experiments, as sentence embeddings are known to be somewhat imprecise at small scales (see Appendix B.4).

**Explore puzzle: arXiv abstracts and titles.** We run similar experiments on a larger-scale dataset of paper titles and abstracts from `arXiv.org`. Using the arXiv API (arXiv.org, 2025), we collect 10 abstract-title pairs from each of the 41 different arXiv categories. To minimize the likelihood that these papers appear in GPT-4O's training corpus, we only use papers uploaded after June 2024 (these experiments were done in Sept'24). Each abstract-title pair yields a task, as discussed earlier.

We evaluate GPT-4O as an exploration oracle for these tasks, much like for the Q/A tasks Given an abstract, we prompt GPT-4O to generate $K$ alternative titles, which are then used to instantiate a bandit algorithm. We use the same algorithm (UCB1) and time horizon $T = 1000$. We record the expected time-averaged reward, $\mathtt{rew} := \frac{1}{T} \sum_{t \in [T]} \mu(a_t)$, and compute the average over tasks within the same arXiv category, $\overline{\mathtt{rew}}$(category, $K$). We try "all-at-once" and "one-by-one" prompting.[10]

We visualize our findings in Figure 8 , using the same conventions as Figure 6 and focusing on six arXiv categories.[11] To assess LLM's ability to specialize to a task, we consider a stronger baseline, where the candidate arms are generated by GPT-4O given only the category, *not the abstract*. For the other arXiv categories, we provide a table with $\overline{\mathtt{rew}}$(category, $K$) for $K = 1, 2, 5$ and both prompts (Appendix B). Like for the Q/A tasks, (1) both prompting strategies significantly outperform the baseline, and (2) their performance tends to improve modestly as $K$ increases.

## 4 CONCLUSIONS

Our work adds to (and provides partial explanations for) the growing literature on in-context learning for decision-making tasks. We find that LLMs are useful as *exploration oracles* that propose high-quality candidate actions in large, semantically meaningful action spaces. However, current LLMs are not that good at *exploitation*, particularly in larger or more complex tasks (we trace the "boundary of tractability" in terms of task size/complexity). While we suggest several helpful mitigations, they consistently underperform relative to a simple linear regression, even in inherently non-linear tasks.

*Future work.* LLMs trained to use tools like a calculator may be better at exploitation. However, it is unclear how much this would help in more complex scenarios, e.g., CB tasks with text-based contexts and actions, and which mitigations/prompting techniques would be needed. Second, while "zooming" bandit algorithms do not work for rich text-based action spaces (discussed in Section 3), LLM-based exploration oracles may potentially help. The hope is to "zoom in" entirely in the space of "potentially relevant" actions (as determined by the LLM), rather than in the space of *all* actions.

---

[10]We do not use "encouragement" on these tasks, since it does not help much (if at all) for the Q/A tasks.

[11]General relativity, computer vision, statistics theory, biomolecules, signal processing, general economics.

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

# A APPENDIX FOR SECTION 2: LLMS AS EXPLOITATION ORACLES

## A.1 PROMPTS

In this section we give example prompts for each of our experimental setups.

"Buttons" prompt for the MAB puzzle:

> [SYSTEM] You are in a room with 5 buttons labeled blue, green, red, yellow, purple. Each button is associated with a Bernoulli distribution with a fixed but unknown mean; the means for the buttons could be different. For each button, when you press it, you will get a reward that is sampled from the button's associated distribution. Then you must pick the button with the highest empirical average, which must be exactly one of blue, green, red, yellow, or purple. You must provide your final answer immediately within the tags <Answer>COLOR <Answer>where COLOR is one of blue, green, red, yellow, or purple and with no text explanation.
>
> [USER] The past rewards for each button are:
>
> round 1: blue button had reward 1, green button had reward 1, red button had reward 0, yellow button had reward 1, purple button had reward 0
>
> round 2: blue button had reward 0, green button had reward 1, red button had reward 1, yellow button had reward 1, purple button had reward 0
>
> Which button do you choose? Remember, YOU MUST provide your final answer within the tags <Answer>COLOR <Answer>where COLOR is one of blue, green, red, yellow, or purple and with no text explanation.

"Adverts" prompt for the MAB puzzle:

> [SYSTEM] You are recommendation engine that chooses advertisements to display to users when they visit your webpage. There are 5 advertisements you can choose from, named A, B, C, D, E. When a user visits the webpage you can choose an advertisement to display and you will observe whether the user would have clicked each of the ads. You model this by assuming that each advertisement has a certain click rate and users click on advertisements with their corresponding rates. I will show you the past clicks for each advertisement. Then you must pick the advertisement with the highest empirical click rate, which must be exactly one of A, B, C, D, or E. You must provide your final answer immediately and with no text explanation. within the tags <Answer>ADVERTISEMENT <Answer>where ADVERTISEMENT is one of A, B, C, D, or E.
>
> [USER] The past clicks for each advertisement are:
>
> round 1: advertisement A was clicked, advertisement B was clicked, advertisement C was not clicked, advertisement D was clicked, advertisement E was clicked
>
> round 1: advertisement A was not clicked, advertisement B was clicked, advertisement C was clicked, advertisement D was clicked, advertisement E was not clicked
>
> Which advertisement do you choose? Remember, YOU MUST provide your final answer within the tags <Answer>ADVERTISEMENT <Answer>where ADVERTISEMENT is one of A, B, C, D, or E and with no text explanation.

"Buttons" prompt for the numerical CB puzzle:

> [SYSTEM] You are in a room with a television and 2 buttons labeled blue, green. Each button is associated with a Bernoulli distribution with an unknown mean; the means for the buttons could be different from each other and may depend on the list of numbers shown on the screen (i.e. the context). For each button, when you press it, you will get a reward that is sampled from the button's associated distribution, conditioned on the numbers shown on the television screen. I will show you the past numbers shown on the screen and the corresponding rewards for each button. A new list of numbers will then appear on the screen and you

must pick the next button in order to maximize your reward in this round only, which must be exactly one of blue or green. You must provide your final answer immediately within the tags ¡Answer¿ COLOR ¡/Answer¿ where COLOR is one of blue or green and with no text explanation.

[USER] The past contexts and rewards for each button are:

In round 1, the context was $[0.3, 0.7]$. The blue button had reward 1, the green button had reward 1

In round 2, the context was $[0.4, 0.6]$. The blue button had reward 0, the green button had reward 1

Which button do you choose? Remember, YOU MUST provide your final answer within the tags <Answer>COLOR <Answer>where COLOR is one of blue or green and with no text explanation.

Prompt for the text-based CB puzzle:

[SYSTEM] You are in a room with a table and a button. There may also be other objects in the room, which I will tell you about. You must then take one of the following actions: "pet animal", "leave room", "use tool", "eat food", "press button", after which you will receive some reward. The reward you receive is a random function of both the action you take and the information you receive about the objects in the room and time of day. Your goal is to maximize the expected reward you receive. I will show you the past history of play over 2 rounds. For each round, I will show you the state of the room and the corresponding rewards for each action. I will then tell you the current state of the room, and you must pick the next action in order to maximize your reward in this round only, which must be exactly one of "pet animal", "leave room", "use tool", "eat food", or "press button". Look for patterns in the data and try to estimate the reward of each action, given the information at your disposal. You must provide your final answer immediately within the tags <Answer>ACTION <Answer>where ACTION is one of "pet animal", "leave room", "use tool", "eat food", or "press button" and with no text explanation.

[USER] The past observations and outcomes for each action are:

Round 1 had context time of day: morning, animal: bear, table item: chest, tool: key, food: apple, button color: red. "pet animal" had reward 0, "leave room" had reward 1, "use tool" had reward 1, "eat food" had reward 0, press button had reward 0

Round 2 had context time of day: afternoon, animal: cat, table item: card, tool: hammer, food: cake, button color: orange. "pet animal" had reward 1, "leave room" had reward 0, "use tool" had reward 0, "eat food" had reward 1, press button had reward 0

The current state of the room is time of day: evening, animal: bear, table item: envelope, tool: key, food: nut, button color: red.

Which action do you choose? Remember, you must provide your final answer immediately within the tags <Answer>ACTION <Answer>where ACTION is one of "pet animal", "leave room", "use tool", "eat food", or "press button" and with no text explanation.

## A.2 ADDITIONAL MAB FIGURES

See Figure 9, Figure 10, Figure 11 for additional results in our MAB exploit puzzle.

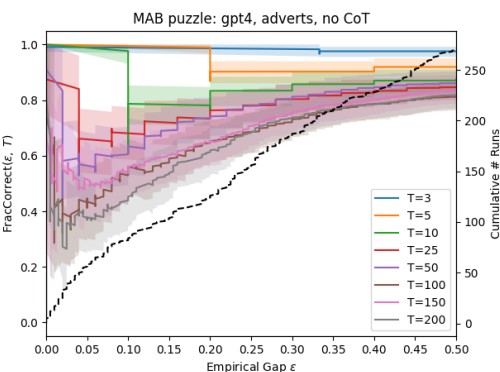

Figure 9: Cumulative fraction correct for GPT-4 in the MAB adverts puzzle.

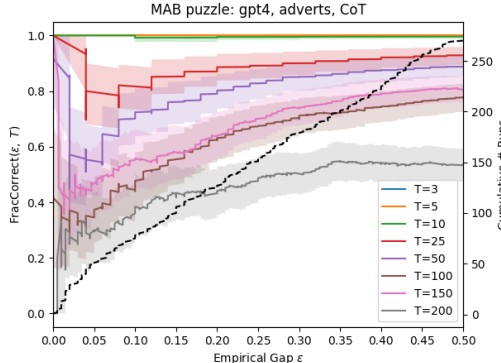

Figure 10: Cumulative fraction correct for GPT-4 with chain-of-thought reasoning in the MAB adverts puzzle.

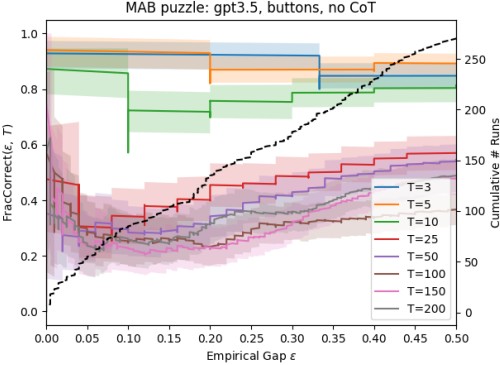

Figure 11: Cumulative fraction correct for GPT-3.5 in the MAB buttons puzzle.

A.3    ADDITIONAL DETAILS FOR TEXT-BASED CB PUZZLES

Each context contains a time of day (belonging to {morning, afternoon, evening, night}), an animal ({bear, dog, cat, None}), a tool ({key, letter opener, hammer, None}), a food item ({cake, apple, nut, None}), and a button with a particular color ({red, orange, yellow, green}). The actions in each round are "pet animal", "leave room", "use tool", "eat food", and "press button".

We experimented with two reward functions: an "easy" reward function, where the expected rewards for each action are as follows:

- The expected reward of petting the animal is 0.01 if the animal is a bear, 0.7 if the animal is a dog, and 0.4 if the animal is a cat. Otherwise, the expected reward if 0.5.

- The expected reward for leaving the room is always 0.5.

- The expected reward for using the tool is 0.75 if it is a key, 0.6 if it is a letter opener, 0.45 if it is a hammer, and 0.2 otherwise.

- The expected reward for eating food is 0.8 if it is cake, 0.6 if it is an apple, 0.2 if it is a nut, and 0.3 otherwise.

- The expected reward for pressing the button is 0.89 if it is green, 0.62 if it is yellow, 0.39 if it is orange, and 0.27 if it is red.

Our results under this reward function are summarized in Figure 12. We used hamming distance to implement our mitigations. Note that in higher-dimensional settings, distance in an embedding space may be used.

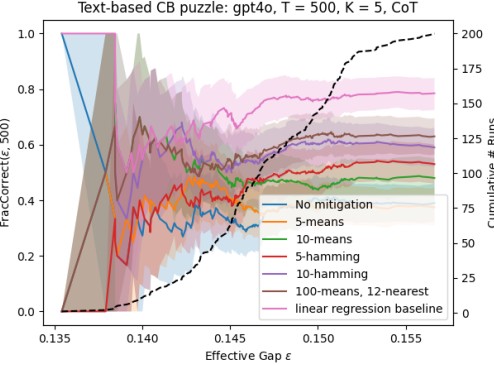

Figure 12: Performance of GPT-4O with mitigations on the words CB puzzle with "easy" rewards.

The reward function we use in the main body is more complicated, and is detailed below:

- The expected reward for petting the animal is 0.01 if it is a bear, 0.7 if it is a dog, 0.3 if it is a cat and the time of day is morning or afternoon, 0.7 if it is a cat and the time of day is evening or night, and otherwise 0.5.

- The expected reward for leaving the room is always 0.5

- If the animal is a bear, the expected reward for using the tool is 0.1. Otherwise, if the tool is a key and the table item is a chest, the expected reward is 0.9. Otherwise, it is 0.4

- If the animal is a bear, the expected reward for eating food is always 0.5. Otherwise, the expected reward is 0.8 for cake, 0.6 for an apple, 0.2 for a nut, and 0.5 otherwise.

- If the animal is a bear, the expected reward for pressing the button is 0.1. Otherwise if the button is green and the time of day is morning, or the button is yellow and the time of day is afternoon, or the button is orange and the time of day is evening, or the button is red and the time of day is night, then the expected reward is 0.9. In all other cases, the expected reward is 0.25.

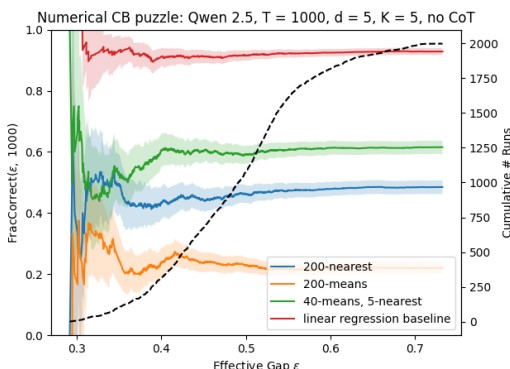

Figure 13: Results for QWEN-2.5 in the setting of Figure 4. Only mitigations are shown, as the unmitigated history is too long to fit in the context window.

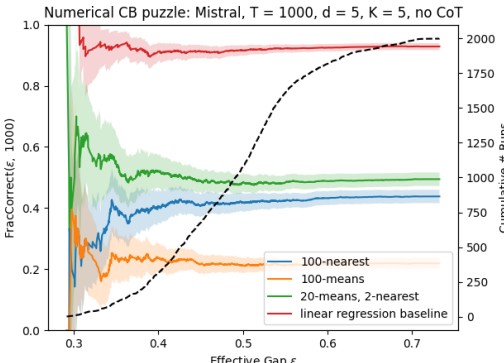

Figure 14: Results for MISTRAL-7B in the setting of Figure 4. Only mitigations are shown, as the unmitigated history is too long to fit in the context window.

### A.4 ROBUSTNESS CHECK WITH OTHER MODELS

In this appendix, we include results for Qwen2.5-7B-Instruct (QWEN-2.5), Gemma 3 12B (GEMMA-3), and Mistral-7B-Instruct-v0.3 (MISTRAL-7B) on our exploit puzzles. Due to the time and costs associated with running our exploit puzzles, we prioritized (1) getting at least one additional set of results for each of our main puzzles (using QWEN-2.5) and (2) getting results for all of our models in our word-based puzzles. Our findings are the same across all models.

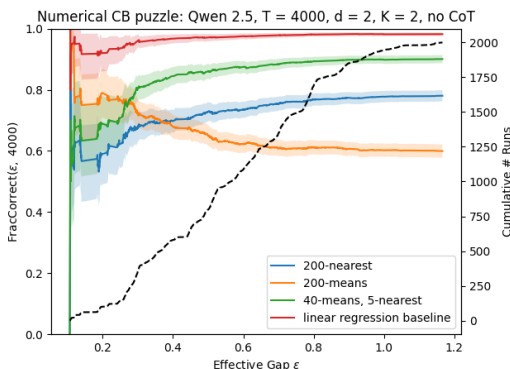

Figure 15: Results for QWEN-2.5 in the setting of Figure 3. Only mitigations are shown, as the unmitigated history is too long to fit in the context window.

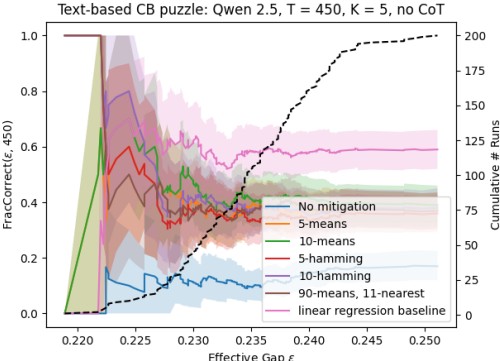

Figure 16: Results for QWEN-2.5 in the setting of Figure 5 (right). We use a slightly smaller history length ($T = 450$ instead of $T = 500$) so that the full history can fit in the context window.

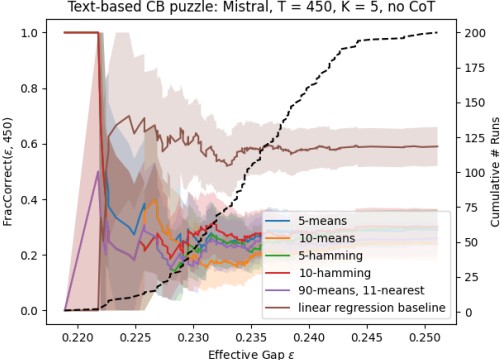

Figure 17: Results for MISTRAL-7B in the setting of Figure 16. Only mitigations are shown, as the full history could not fit in the context window.

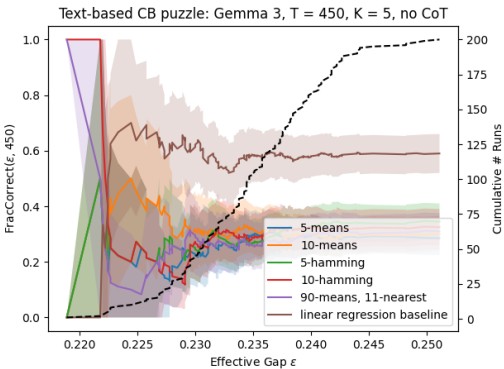

Figure 18: Results for GEMMA-3 in the setting of Figure 16. Only mitigations are shown, as the full history could not fit in the context window.

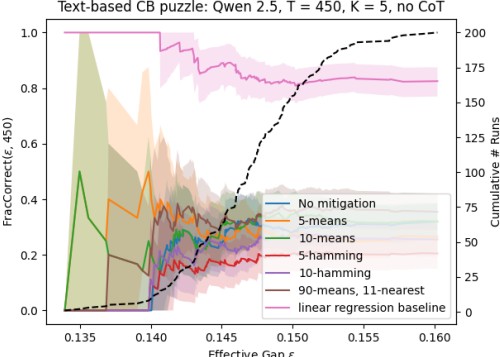

Figure 19: Results for QWEN-2.5 in the setting of Figure 12. We use a slightly smaller history length ($T = 450$ instead of $T = 500$) so that the full history can fit in the context window.

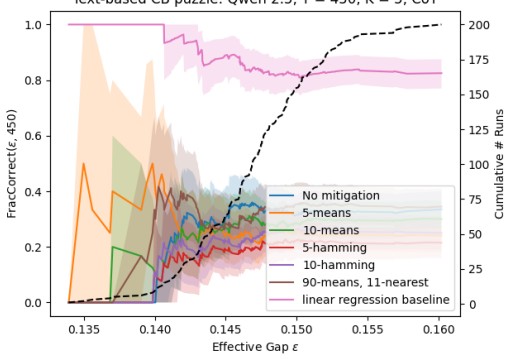

Figure 20: Results for QWEN-2.5 in the setting of Figure 12, using chain-of-thought prompting. We use a slightly smaller history length ($T = 450$ instead of $T = 500$) so that the full history can fit in the context window.

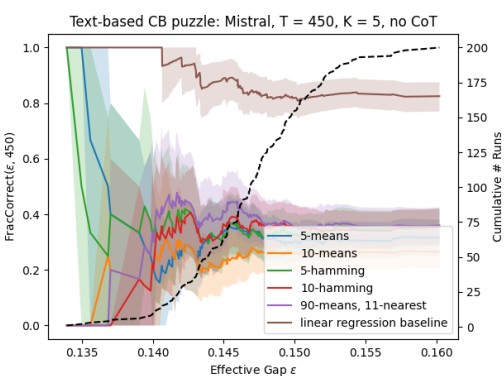

Figure 21: Results for MISTRAL-7B in the setting of Figure 19. Only mitigations are shown, as the full history could not fit in the context window.

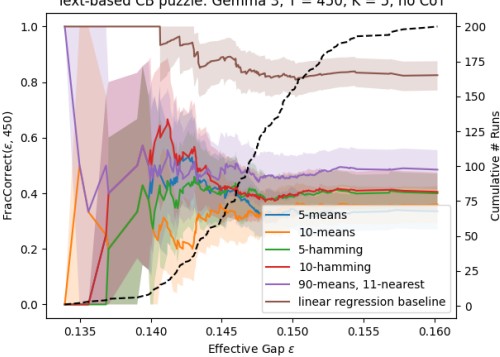

Figure 22: Results for GEMMA-3 in the setting of Figure 19. Only mitigations are shown, as the full history could not fit in the context window.

## A.5 SAMPLE REASONING TRACES

In this section, we provide the raw thinking tokens for DEEPSEEK-R1-DISTILL-QWEN on some sample exploit puzzles.

Sample thinking tokens for the "buttons" numerical CB puzzle:

> Alright, I'm trying to figure out which button to press next based on the context given. The context is a list of five numbers: [-0.671, 0.267, 0.415, 0.284, -0.775]. I need to choose between blue, green, red, yellow, or purple to maximize the reward.
>
> First, I'll look at the past data to see if there's any pattern or relationship between the context and the rewards for each button. Each button has been pressed multiple times, and each time, a reward was given. My goal is to estimate which button is likely to give the highest reward for the current context.
>
> I notice that each context is a 5-dimensional vector, and each button's reward seems to vary based on these values. Maybe I can find a way to predict the reward for each button given the context.
>
> One approach is to look for correlations between the context and the rewards for each button. For example, if a certain context value tends to be high when a button's reward is also high, that button might be a good choice.
>
> Let me list out the context values and the rewards for each button in the past rounds. However, since there are 50 rounds, that's a lot of data to process manually. Instead, I'll try to identify if any of the context dimensions have a consistent positive or negative impact on the rewards for each button.
>
> Looking at the current context: [-0.671, 0.267, 0.415, 0.284, -0.775]. I'll note the positions:
>
> -0.671 0.267 0.415 0.284 -0.775 Now, I'll think about each button's performance in similar contexts.
>
> For example, in round 1, the context was [-0.149, -0.038, -0.854, -0.664, 0.534], and red had a high reward. In round 2, context [0.357, 0.022, 0.079, 0.118, -0.262], green had a high reward. Round 3: context [-0.155, 0.23, -0.274, 0.333, 0.837], red had a high reward. Round 4: context [-0.242, -0.94, 0.664, -0.708, -0.624], green had a high reward.
>
> Hmm, it's a bit scattered. Maybe I should look for buttons that consistently perform well when certain context values are present.
>
> Looking at the current context, the first and fifth values are negative, while the second, third, and fourth are positive. I'll check if any buttons tend to perform well when the first and fifth are negative, and the middle three are positive.
>
> In round 1: first negative, fifth positive. Red had high reward.
>
> Round 2: first positive, fifth negative. Green had high reward.
>
> Round 3: first negative, fifth positive. Red had high reward.
>
> Round 4: first negative, fifth negative. Green had high reward.
>
> Round 5: first negative, fifth negative. Green had high reward.
>
> Round 6: first negative, fifth positive. Red had a moderate reward.
>
> Round 7: first positive, fifth positive. Red had a moderate reward.
>
> Round 8: first positive, fifth negative. Green had high reward.
>
> Round 9: first negative, fifth positive. Red had high reward.
>
> Round 10: first negative, fifth positive. Red had a moderate reward.
>
> Round 11: first negative, fifth positive. Green had high reward.
>
> Round 12: first positive, fifth negative. Green had moderate reward.
>
> Round 13: first positive, fifth negative. Green had moderate reward.
>
> Round 14: first positive, fifth positive. Green had negative reward, red had high.
>
> Round 15: first negative, fifth positive. Green had negative, red had moderate.
>
> Round 16: first negative, fifth negative. Green had high reward.

Round 17: first positive, fifth negative. Yellow and purple had high rewards.

Round 18: first negative, fifth positive. Green and red had high rewards.

Round 19: first negative, fifth positive. Green had negative, red had negative, purple had moderate.

Round 20: first negative, fifth positive. Green and red had high rewards.

Round 21: first negative, fifth negative. Green had high reward.

Round 22: first positive, fifth positive. Green had negative, red had moderate.

Round 23: first positive, fifth positive. Yellow had high reward.

Round 24: first negative, fifth negative. Green had high reward.

Round 25: first positive, fifth positive. Yellow had moderate reward.

Round 26: first negative, fifth positive. Green and red had high rewards.

Round 27: first positive, fifth positive. Green had negative, red had moderate.

Round 28: first positive, fifth negative. Green had high reward.

Round 29: first positive, fifth positive. Yellow had high reward.

Round 30: first positive, fifth positive. Red had moderate reward.

Round 31: first negative, fifth negative. Green had high reward.

Round 32: first negative, fifth positive. Purple had high reward.

Round 33: first positive, fifth positive. Yellow had high reward.

Round 34: first negative, fifth positive. Green had moderate reward.

Round 35: first positive, fifth positive. Purple had high reward.

Round 36: first positive, fifth positive. Red had high reward.

Round 37: first negative, fifth positive. Red had moderate reward.

Round 38: first negative, fifth negative. Red had negative, yellow had moderate.

Round 39: first negative, fifth positive. Purple had moderate reward.

Round 40: first positive, fifth positive. Red had high reward.

Round 41: first positive, fifth negative. Yellow had high reward.

Round 42: first negative, fifth positive. Green had negative, red had moderate.

Round 43: first negative, fifth positive. Green had moderate, purple had high.

Round 44: first positive, fifth positive. Green and red had high rewards.

Round 45: first positive, fifth positive. Yellow and purple had high rewards.

Round 46: first positive, fifth negative. Green had high reward.

Round 47: first positive, fifth positive. Yellow had high reward.

Round 48: first negative, fifth positive. Green had negative, purple had moderate.

Round 49: first positive, fifth negative. Green and red had high rewards.

Round 50: first positive, fifth positive. Yellow had high reward.

From this, it seems that when the first context value is negative and the fifth is positive, red and green buttons often have high rewards. When the first is positive and the fifth is negative, green and sometimes yellow perform well. When both first and fifth are negative, green tends to do well. When both are positive, it's mixed, but red, green, yellow, and purple have had high rewards.

In the current context, the first is negative (-0.671), and the fifth is negative (-0.775). So both first and fifth are negative. From past data, when both are negative, green often has a high reward. For example, round 4, 5, 16, 24, 31, etc., green performed well in such contexts.

Therefore, I think pressing the green button would be the best choice for this context to maximize the reward.

Sample thinking tokens for the text-based CB puzzle:

Alright, let's try to figure out the best action to take in the current state. The current observation is: time of day is night, animal is bear, table item is card, tool is key, food is apple, and button color is red. I need to choose between pet animal, leave room, use tool, eat food, or press button, aiming for the highest reward.

First, I'll look through the past 50 rounds to find similar states and see what actions yielded the highest rewards. Let's break down each possible action and see if there's a pattern.

Pet Animal: Looking at previous rounds where the animal was a bear, especially at night. In Round 3, the animal was a bear, time was afternoon, and petting gave 0.01. In Round 5, same animal and time, 0.01. Round 19 had no animal, but others with bear had low rewards. So, probably not a good choice.

Leave Room: This action always gives 0.5, regardless of other factors. It's a safe choice but might not be the highest reward.

Use Tool: The tool here is a key. Looking at past rounds where the tool was a key. For example, Round 4 had a key and use tool gave 0.75. Round 10 also had a key and use tool gave 0.75. So, when the tool is a key, using it seems to give a good reward. But wait, in the current state, the tool is a key, but what about the other items? The table item is a card, and food is apple. Maybe the presence of a card affects it? Not sure, but the key seems to be a good tool.

Eat Food: The food here is apple. Looking at past rounds with apple as food. For example, Round 2 had apple and eat food gave 0.8. Round 5 had apple and 0.6. Round 10 had apple and 0.6. So, eating apple can give high rewards, but it varies. It might be a good option, but not as consistent as other actions.

Press Button: The button color is red. Looking at past rounds with red button. Round 9 had red button and press button gave 0.27. Round 41 had red and 0.27. So, pressing the button when it's red doesn't seem good, as the reward is low.

Now, considering all this, the highest rewards seem to come from using the tool (key) and eating food (apple). Between these two, using the tool has consistently given higher rewards when the tool is a key. However, I should check if there are any specific interactions between the tool and other items. The table item is a card, which I haven't seen much in the data, but using the tool with a key might still be beneficial.

Alternatively, eating the apple could give a high reward, but it's not as consistent. Since using the tool with a key has given 0.75 in previous rounds, which is higher than the 0.8 from eating, but wait, in some cases, eating apple gave 0.8, which is higher. Hmm, but I need to consider the current state's specifics.

Wait, in the current state, the tool is a key, and the table item is a card. Maybe using the key on the card? But I don't have information on that interaction. However, in past rounds where the tool was a key, using it gave high rewards regardless of other items. So, perhaps using the tool is the better choice here.

Alternatively, eating the apple could give a good reward, but I'm not sure if it's higher than using the tool. Given that using the tool has given higher rewards in similar situations, I think it's safer to go with "use tool" to maximize the reward.

## B APPENDIX FOR SECTION 3: LLMs AS EXPLORATION ORACLES

### B.1 PROMPTS

In this section we give example prompts for each of our experimental setups.

"All-at-once" prompt with encouragement for open-ended question puzzle:

> [SYSTEM] I will give you an open-ended question. Come up with 5 different candidate answers. Reply only with the 5 candidate answers, and put each candidate answer on a separate line. Each answer should only be a few words, skipping any introductory phrasing and going straight to the essence. Try to come up with answers that are very different in spirit from one another.
> [USER] Here is the question: "What is the purpose of art?"

"One-by-one" prompt without encouragement for arXiv puzzle:

> [SYSTEM] I will give you an abstract and some candidate titles for a paper. Come up with a new candidate title that is relevant to the abstract, but different from the other candidate titles. Reply only with the candidate title.
> [USER] Here is the abstract: {abstract goes here}
> Here are the other candidate titles: {previous suggestions go here}

### B.2 DATASETS

Our open-ended question dataset consists of the following 10 questions and the corresponding "ground-truth" answers.

Questions:

0. What is the meaning of freedom?
1. How should we define success?
2. What is the role of technology in society?
3. What is the nature of reality?
4. What is the purpose of art?
5. What does it mean to live a fulfilling life?
6. How do cultural differences shape our understanding of morality?
7. What is the relationship between happiness and wealth?
8. How can we balance individuality and community in modern society?
9. What is the role of education in personal and societal growth?

Answers:

0. Freedom is an illusion shaped by societal norms and external influences.
1. Success should be defined as contributing to the greater good rather than personal achievement.
2. Technology disrupts the natural balance of society and often creates more problems than it solves.
3. Reality is subjective, varying entirely based on individual perception and experience.
4. The purpose of art is to challenge conventions and disrupt established ideas.
5. Fulfillment comes from embracing discomfort.
6. Cultural differences create moral superiority.
7. Wealth detracts from true happiness.
8. Individuality thrives when shaped by community.

9. Education's purpose is to challenge authority.

Here is the list of paper titles we used in our arXiv dataset, along with their corresponding categories:

gr-qc

1. There is more to the de Sitter horizon than just the area

2. Mitigating cosmic variance in the Hellings-Downs curve: a Cosmic Microwave Background analogy

3. Calabi-Yau Feynman integrals in gravity: $\varepsilon$-factorized form for apparent singularities

4. QG from SymQRG: AdS$_3$/CFT$_2$ Correspondence as Topological Symmetry-Preserving Quantum RG Flow

5. Black hole solutions in theories of supergravity

6. Horndeski in motion

7. Wormholes from beyond

8. Regularizing the Pulsar Timing Array likelihood: A path towards Fourier Space

9. Solutions to the mode equation for a quantized massless scalar field outside a black hole that forms from the collapse of a null shell: Late-time behaviors and computation of the stress-energy tensor

10. Gravitational waves from regular black holes in extreme mass-ratio inspirals

hep-ex

1. Observation of the $K^+ \to \pi^+ \nu \bar{\nu}$ decay and measurement of its branching ratio

2. Test of lepton flavour universality in $W$-boson decays into electrons and $\tau$-leptons using $pp$ collisions at $\sqrt{s} = 13$ TeV with the ATLAS detector

3. Searching for neutrino self-interactions at future muon colliders

4. Quantum Decoherence at ESSnuSB Experiment

5. Test of lepton flavour universality with $B^+ \to K^+ \pi^+ \pi^- \ell^+ \ell^-$ decays

6. Cross-section measurements for the production of a $W$-boson in association with high-transverse-momentum jets in $pp$ collisions at $\sqrt{s}= 13$ TeV with the ATLAS detector

7. Charmful two-body $\Omega_b$ decays in the light-front quark model

8. Observation of a spectral hardening in cosmic ray boron spectrum with the DAMPE space mission

9. New BaBar studies of high-order radiation and the new landscape of data-driven HVP predictions of the muon g-2

10. Toponium: the smallest bound state and simplest hadron in quantum mechanics

hep-lat

1. Quantum sampling on a quantum annealer for large volumes in the strong coupling limit for gauge group U(3)

2. Phase diagram of Rydberg atoms in a two-leg rectangular ladder

3. Graph Attention Hamiltonian Neural Networks: A Lattice System Analysis Model Based on Structural Learning

4. What do we know about the confinement mechanism?

5. Designing weight regularizations based on Lefschetz thimbles to stabilize complex Langevin

6. Likelihood of a zero in the proton elastic electric form factor

7. Real-Time Simulation of Asymmetry Generation in Fermion-Bubble Collisions

8. Investigating SU(3) with Nf=8 fundamental fermions at strong renormalized coupling

9. The determination of potential scales in 2+1 flavor QCD

10. Towards the phase diagram of fermions coupled with $SO(3)$ quantum links in $(2+1)$-D

hep-ph

1. Predictions for dimuon production in high-energy neutrino-proton collisions using the color dipole model

2. Extrapolating Jet Radiation with Autoregressive Transformers

3. Accurate Surrogate Amplitudes with Calibrated Uncertainties

4. Calabi-Yau Feynman integrals in gravity: $\varepsilon$-factorized form for apparent singularities

5. The causal structure of the quark propagator

6. Fuzzy Axions and Associated Relics

7. Non-Radial Oscillation Modes in Hybrid Stars with Hyperons and Delta Baryons: Full General Relativity Formalism vs. Cowling Approximation

8. Evidence for the Sombrero Galaxy as an Accelerator of the Highest-Energy Cosmic Rays

9. The cosmic history of Primordial Black Hole accretion and its uncertainties

10. Searching for neutrino self-interactions at future muon colliders

hep-th

1. There is more to the de Sitter horizon than just the area

2. Calabi-Yau Feynman integrals in gravity: $\varepsilon$-factorized form for apparent singularities

3. QG from SymQRG: AdS$_3$/CFT$_2$ Correspondence as Topological Symmetry-Preserving Quantum RG Flow

4. Geometrically constrained localized configurations engendering non-topological profile

5. The causal structure of the quark propagator

6. Entanglement Hamiltonian and orthogonal polynomials

7. Black hole solutions in theories of supergravity

8. Fuzzy Axions and Associated Relics

9. Celestial Mellin Amplitudes

10. Evidence for the Sombrero Galaxy as an Accelerator of the Highest-Energy Cosmic Rays

math-ph

1. QG from SymQRG: AdS$_3$/CFT$_2$ Correspondence as Topological Symmetry-Preserving Quantum RG Flow

2. Entanglement Hamiltonian and orthogonal polynomials

3. Fermi's golden rule in tunneling models with quantum waveguides perturbed by Kato class measures

4. Semiclassical measure of the propagation between two topological insulators

5. On the Protection Against Noise for Measurement-Based Quantum Computation

6. Calculating Spectra by Sequential High-Pass Filtering

7. Validity of the stochastic Landau approximation for super-pattern forming systems with a spatial 1:3 resonance

8. Multi-component Hamiltonian difference operators

9. Emptiness Instanton in Quantum Polytropic Gas

10. Unitary $n$-correlations with restricted support in random matrix theory

nucl-ex

1. The evidence of $N = 16$ shell closure and $\beta$-delayed neutron emission from $\wedge^{25}$F

2. Isotopic Transparency in Central Xe+Sn Collisions at 100 MeV/nucleon

3. Detecting the Coupling of Axion Dark Matter to Neutron Spins at Spallation Sources via Rabi Oscillation

4. Likelihood of a zero in the proton elastic electric form factor

5. Nuclear structure and direct reaction studies in particle-$\gamma$ coincidence experiments at the FSU John D. Fox Superconducting Linear Accelerator Laboratory

6. Bottomonium-like states in proton collisions: Fragmentation and resummation

7. Towards a foundation model for heavy-ion collision experiments through point cloud diffusion

8. Study of the energy spectrum of alpha particles in an experiment on irradiation of a boron target with a proton beam at the Prometheus accelerator

9. Staking out the Proton Drip-Line of Thulium at the N=82 Shell Closure

10. Measurements of global and local spin polarization of $\wedge$ and $\bar{\wedge}$ in Au+Au collisions from the RHIC Beam Energy Scan

nucl-th

1. The causal structure of the quark propagator

2. Non-Radial Oscillation Modes in Hybrid Stars with Hyperons and Delta Baryons: Full General Relativity Formalism vs. Cowling Approximation

3. Isotopic Transparency in Central Xe+Sn Collisions at 100 MeV/nucleon

4. Quantum-Corrected Holographic Wilson Loop Correlators and Confinement

5. Dynamics of Hot QCD Matter 2024 – Bulk Properties

6. Spurious Isospin Breaking in the In-medium Similarity Renormalization Group

7. Likelihood of a zero in the proton elastic electric form factor

8. Born-Oppenheimer Renormalization group for High Energy Scattering: the Modified BFKL, or where did it all go?

9. Nuclear structure and direct reaction studies in particle-$\gamma$ coincidence experiments at the FSU John D. Fox Superconducting Linear Accelerator Laboratory

10. Bottomonium-like states in proton collisions: Fragmentation and resummation

quant-ph

1. Noisy initial-state qubit-channel metrology with additional undesirable noisy evolution

2. The State Preparation of Multivariate Normal Distributions using Tree Tensor Network

3. QG from SymQRG: AdS$_3$/CFT$_2$ Correspondence as Topological Symmetry-Preserving Quantum RG Flow

4. Entanglement Hamiltonian and orthogonal polynomials

5. Learning interactions between Rydberg atoms

6. The isoholonomic inequality and tight implementations of holonomic quantum gates

7. Fermi's golden rule in tunneling models with quantum waveguides perturbed by Kato class measures

8. Temporal evolution of a forced optomechanical system with linear and quadratic field – mechanical oscillator couplings

9. Control of a Josephson Digital Phase Detector via an SFQ-based Flux Bias Driver

10. Commentary on the decomposition of universal multiport interferometers: how it works in practice

cs.AI

1. MaxInfoRL: Boosting exploration in reinforcement learning through information gain maximization

2. SepLLM: Accelerate Large Language Models by Compressing One Segment into One Separator

3. Stabilizing Reinforcement Learning in Differentiable Multiphysics Simulation

4. Revelations: A Decidable Class of POMDPs with Omega-Regular Objectives

5. Artificial Intelligence in Traffic Systems

6. The Impact of AI Assistance on Radiology Reporting: A Pilot Study Using Simulated AI Draft Reports

7. Can LLM Prompting Serve as a Proxy for Static Analysis in Vulnerability Detection

8. FSFM: A Generalizable Face Security Foundation Model via Self-Supervised Facial Representation Learning

9. Learning to Navigate in Mazes with Novel Layouts using Abstract Top-down Maps

10. SpeechPrune: Context-aware Token Pruning for Speech Information Retrieval

cs.CL

1. SepLLM: Accelerate Large Language Models by Compressing One Segment into One Separator

2. Making FETCH! Happen: Finding Emergent Dog Whistles Through Common Habitats

3. Semi-automated analysis of audio-recorded lessons: The case of teachers' engaging messages

4. Virtual Agent-Based Communication Skills Training to Facilitate Health Persuasion Among Peers

5. How Private are Language Models in Abstractive Summarization?

6. Can LLM Prompting Serve as a Proxy for Static Analysis in Vulnerability Detection

7. SpeechPrune: Context-aware Token Pruning for Speech Information Retrieval

8. The Open Source Advantage in Large Language Models (LLMs)

9. LLM-RG4: Flexible and Factual Radiology Report Generation across Diverse Input Contexts

10. ExecRepoBench: Multi-level Executable Code Completion Evaluation

cs.CV

1. PanSplat: 4K Panorama Synthesis with Feed-Forward Gaussian Splatting

2. Causal Diffusion Transformers for Generative Modeling

3. CAP4D: Creating Animatable 4D Portrait Avatars with Morphable Multi-View Diffusion Models

4. Wonderland: Navigating 3D Scenes from a Single Image

5. Stabilizing Reinforcement Learning in Differentiable Multiphysics Simulation

6. Instruction-based Image Manipulation by Watching How Things Move

7. IDArb: Intrinsic Decomposition for Arbitrary Number of Input Views and Illuminations

8. UniLoc: Towards Universal Place Recognition Using Any Single Modality

9. CPath-Omni: A Unified Multimodal Foundation Model for Patch and Whole Slide Image Analysis in Computational Pathology

10. CG-Bench: Clue-grounded Question Answering Benchmark for Long Video Understanding

cs.LG

1. MaxInfoRL: Boosting exploration in reinforcement learning through information gain maximization

2. SepLLM: Accelerate Large Language Models by Compressing One Segment into One Separator

3. No More Tuning: Prioritized Multi-Task Learning with Lagrangian Differential Multiplier Methods

4. Stabilizing Reinforcement Learning in Differentiable Multiphysics Simulation

5. Extrapolating Jet Radiation with Autoregressive Transformers

6. Bilevel Learning with Inexact Stochastic Gradients

7. LLMs for Cold-Start Cutting Plane Separator Configuration

8. LeARN: Learnable and Adaptive Representations for Nonlinear Dynamics in System Identification

9. Thermodynamics-informed graph neural networks for real-time simulation of digital human twins

10. Memory-Reduced Meta-Learning with Guaranteed Convergence

cs.NE

1. Deep-learning-based identification of individual motion characteristics from upper-limb trajectories towards disorder stage evaluation

2. Speeding Up the NSGA-II With a Simple Tie-Breaking Rule

3. Optimal Gradient Checkpointing for Sparse and Recurrent Architectures using Off-Chip Memory

4. Runtime Analysis for Multi-Objective Evolutionary Algorithms in Unbounded Integer Spaces

5. Theoretical Analysis of Quality Diversity Algorithms for a Classical Path Planning Problem

6. Populating cellular metamaterials on the extrema of attainable elasticity through neuroevolution

7. Deployment Pipeline from Rockpool to Xylo for Edge Computing

8. Interlocking-free Selective Rationalization Through Genetic-based Learning

9. EVOS: Efficient Implicit Neural Training via EVOlutionary Selector

10. Brain-inspired Chaotic Graph Backpropagation for Large-scale Combinatorial Optimization

cs.RO

1. MaxInfoRL: Boosting exploration in reinforcement learning through information gain maximization

2. Stabilizing Reinforcement Learning in Differentiable Multiphysics Simulation

3. LeARN: Learnable and Adaptive Representations for Nonlinear Dynamics in System Identification

4. Backstepping Control of Tendon-Driven Continuum Robots in Large Deflections Using the Cosserat Rod Model

5. Learning to Navigate in Mazes with Novel Layouts using Abstract Top-down Maps

6. Emma-X: An Embodied Multimodal Action Model with Grounded Chain of Thought and Look-ahead Spatial Reasoning

7. Lightweight Decentralized Neural Network-Based Strategies for Multi-Robot Patrolling

8. Learning Human-Aware Robot Policies for Adaptive Assistance

9. Hardware-in-the-loop Simulation Testbed for Geomagnetic Navigation

10. Sonar-based Deep Learning in Underwater Robotics: Overview, Robustness and Challenges

cs.IT

1. Codes from $A_m$-invariant polynomials

2. BA-BFL: Barycentric Aggregation for Bayesian Federated Learning

3. Capacity of Hierarchical Secure Coded Gradient Aggregation with Straggling Communication Links

4. Wireless Environmental Information Theory: A New Paradigm towards 6G Online and Proactive Environment Intelligence Communication

5. Quantum search in a dictionary based on fingerprinting-hashing

6. Identification Over Binary Noisy Permutation Channels

7. Iterative Detection and Decoding for Clustered Cell-Free Massive MIMO Networks

8. Structured Sampling for Robust Euclidean Distance Geometry

9. Study of Iterative Detection and Decoding for Multiuser Systems and MMSE Refinements with Active or Passive RIS

10. Shannon information and integrated information: message and meaning

cs.CR

1. Can LLM Prompting Serve as a Proxy for Static Analysis in Vulnerability Detection

2. Efficient Layered New Bit-Flipping QC-MDPC Decoder for BIKE Post-Quantum Cryptography

3. But Can You Use It? Design Recommendations for Differentially Private Interactive Systems

4. Efficiently Achieving Secure Model Training and Secure Aggregation to Ensure Bidirectional Privacy-Preservation in Federated Learning

5. On Large Language Models in Mission-Critical IT Governance: Are We Ready Yet?

6. Just a Simple Transformation is Enough for Data Protection in Vertical Federated Learning

7. SeSeMI: Secure Serverless Model Inference on Sensitive Data

8. DB-PAISA: Discovery-Based Privacy-Agile IoT Sensing+Actuation

9. OTA-Key: Over the Air Key Management for Flexible and Reliable IoT Device Provision

10. Android App Feature Extraction: A review of approaches for malware and app similarity detection

cs.DS

1. Approximating the Top Eigenvector in Random Order Streams

2. Witty: An Efficient Solver for Computing Minimum-Size Decision Trees

3. Adaptive Manipulation for Coalitions in Knockout Tournaments

4. Counting Butterflies over Streaming Bipartite Graphs with Duplicate Edges

5. Quantum search in a dictionary based on fingerprinting-hashing

6. Regularized Dikin Walks for Sampling Truncated Logconcave Measures, Mixed Isoperimetry and Beyond Worst-Case Analysis

7. Proportionally Fair Matching via Randomized Rounding

8. Logarithmic Positional Partition Interval Encoding

9. New results for the detection of bicliques

10. Deterministic Even-Cycle Detection in Broadcast CONGEST

cs.HC

1. Virtual Agent-Based Communication Skills Training to Facilitate Health Persuasion Among Peers

2. The Impact of AI Assistance on Radiology Reporting: A Pilot Study Using Simulated AI Draft Reports

3. Combining Large Language Models with Tutoring System Intelligence: A Case Study in Caregiver Homework Support

4. But Can You Use It? Design Recommendations for Differentially Private Interactive Systems

5. LLMs Can Simulate Standardized Patients via Agent Coevolution

6. LLM-DaaS: LLM-driven Drone-as-a-Service Operations from Text User Requests

7. Private Yet Social: How LLM Chatbots Support and Challenge Eating Disorder Recovery

8. Task-Based Role-Playing VR Game for Supporting Intellectual Disability Therapies

9. Privacy-Preserving Brain-Computer Interfaces: A Systematic Review

10. Accurate, Robust and Privacy-Preserving Brain-Computer Interface Decoding

math.AG

1. Geometry of 3-dimensional del Pezzo fibrations in positive characteristic

2. The Mordell-Schinzel conjecture for cubic diophantine equations

3. The many faces of a logarithmic scheme

4. Lorentzian polynomials and the incidence geometry of tropical linear spaces

5. $p$-adic Local Langlands Correspondence

6. Real del Pezzo surfaces without points

7. Linearization problem for finite subgroups of the plane Cremona group

8. Groupes de monodromie finie des variétés abéliennes

9. Duality for Arithmetic $p$-adic Pro-étale Cohomology of Analytic Spaces

10. The external activity complex of a pair of matroids

math.AT

1. Digital $n-$Manifolds With Or Without Boundaries

2. Spatiotemporal Persistence Landscapes

3. Simplifications of finite spaces equipped with sheaves

4. Rational homotopy theory of operad modules through colored operads

5. Algebraic Topology Without Open Sets: A Net Approach to Homotopy Theory in Limit Spaces

6. The geometry of simplicial distributions on suspension scenarios

7. On the Last Kervaire Invariant Problem

8. Machine Proofs for Adams Differentials and Extension Problems among CW Spectra

9. Finite asymptotic dimension and the coarse assembly map

10. Modeling $(\infty, 1)$-categories with Segal spaces

math.AP

1. Decay estimates for massive Dirac equation in a constant magnetic field

2. Semiclassical measure of the propagation between two topological insulators

3. Convex waves grazing convex obstacles to high order

4. A Note on Hyperbolic Relaxation of the Navier-Stokes-Cahn-Hilliard system for incompressible two-phase flow

5. Positive solutions to general semilinear overdetermined boundary problems

6. Capacitary measures in fractional order Sobolev spaces: Compactness and applications to minimization problems

7. Validity of the stochastic Landau approximation for super-pattern forming systems with a spatial 1:3 resonance

8. Spectral bounds for the operator pencil of an elliptic system in an angle

9. Infinite dimensional invariant tori for nonlinear Schrödinger equations

10. A Serrin-type over-determined problem for Hessian equations in the exterior domain

math.CT

1. Open Condensed Subgroups and Mackey's Formula

2. The Relational Quotient Completion

3. Classification of localizing subcategories along t-structures

4. Categorification of modules and construction of schemes

5. Rational RG flow, extension, and Witt class

6. Intrinsically Correct Sorting in Cubical Agda

7. Single and multi-valued Hilbert-bundle renormings

8. Extended (tri)dendriform algebras, pre-Lie algebras and post-Lie algebras as companion structures of extended Rota-Baxter algebras

9. On The Telescopic Picard Group

10. Enhanced 2-categorical structures, two-dimensional limit sketches and the symmetry of internalisation

math.GR

1. $F$-birestriction monoids in enriched signature

2. Linearization problem for finite subgroups of the plane Cremona group

3. Salter's question on the image of the Burau representation of $B_4$

4. Averaging groups

5. Enumerating Diagonalizable Matrices over $\mathbb{Z}_{p^k}$

6. The scale function for locally compact groups acting on non-positively curved spaces

7. A computational study of certain Weyl modules for type $G_2$ in characteristic 2

8. Left-Invariant Riemannian Distances on Higher-Rank Sol-Type Groups

9. Growth Rate Gap for Stable Subgroups

10. Computing Young's Natural Representations for Generalized Symmetric Groups

math.NT

1. The Mordell-Schinzel conjecture for cubic diophantine equations

2. Simultaneous and multiplicative Diophantine approximation on missing-digit fractals

3. Codes from $A_m$-invariant polynomials

4. Generalised Fermat equation: a survey of solved cases

5. Groupes de monodromie finie des variétés abéliennes

6. Vanishing of Witten zeta function at negative integers

7. Popa's "Recurrent Sequences" and Reciprocity

8. Duality for Arithmetic $p$-adic Pro-étale Cohomology of Analytic Spaces

9. About Eisenstein's Theorem

10. On the packing dimension of weighted singular matrices on fractals

math.OC

1. Bilevel Learning with Inexact Stochastic Gradients

2. Memory-Reduced Meta-Learning with Guaranteed Convergence

3. On Differential Stability of a Class of Convex Optimization Problems

4. Convergence of trust-region algorithms in compact metric spaces

5. Eckstein-Ferris-Pennanen-Robinson duality revisited: paramonotonicity, total Fenchel-Rockallar duality, and the Chambolle-Pock operator

6. Capacitary measures in fractional order Sobolev spaces: Compactness and applications to minimization problems

7. A monotone block coordinate descent method for solving absolute value equations

8. Bivariate rational approximations of the general temperature integral

9. Toward a Unified Theory of Gradient Descent under Generalized Smoothness

10. A particle system approach towards the global well-posedness of master equations for potential mean field games of control

math.ST

1. Optimality of the Right-Invariant Prior

2. The entropic optimal (self-)transport problem: Limit distributions for decreasing regularization with application to score function estimation

3. Causal Invariance Learning via Efficient Optimization of a Nonconvex Objective

4. A partial likelihood approach to tree-based density modeling and its application in Bayesian inference

5. Dual Unscented Kalman Filter Architecture for Sensor Fusion in Water Networks Leak Localization

6. Learning Massive-scale Partial Correlation Networks in Clinical Multi-omics Studies with HP-ACCORD

7. Well-Posedness and Stability of the Stochastic OGTT Model

8. Posterior asymptotics of high-dimensional spiked covariance model with inverse-Wishart prior

9. Model checking for high dimensional generalized linear models based on random projections

10. The Stein-log-Sobolev inequality and the exponential rate of convergence for the continuous Stein variational gradient descent method

q-bio.BM

1. Category-Specific Topological Learning of Metal-Organic Frameworks

2. Applications of Knot Theory for the Improvement of the AlphaFold Protein Database

3. EquiFlow: Equivariant Conditional Flow Matching with Optimal Transport for 3D Molecular Conformation Prediction

4. FlowDock: Geometric Flow Matching for Generative Protein-Ligand Docking and Affinity Prediction

5. NeuralPLexer3: Physio-Realistic Biomolecular Complex Structure Prediction with Flow Models

6. COMET: Benchmark for Comprehensive Biological Multi-omics Evaluation Tasks and Language Models

7. Quadratic unconstrained binary optimization and constraint programming approaches for lattice-based cyclic peptide docking

8. High-dimensional Statistics Applications to Batch Effects in Metabolomics

9. Precise Antigen-Antibody Structure Predictions Enhance Antibody Development with HelixFold-Multimer

10. Sampling-based Continuous Optimization with Coupled Variables for RNA Design

q-bio.GN

1. BarcodeMamba: State Space Models for Biodiversity Analysis

2. VEPerform: a web resource for evaluating the performance of variant effect predictors

3. A robust, scalable K-statistic for quantifying immune cell clustering in spatial proteomics data

4. Can linguists better understand DNA?

5. A Misclassification Network-Based Method for Comparative Genomic Analysis

6. DNA Fragments in Crude Oil Reveals Earth's Hidden History

7. Ancient DNA from 120-Million-Year-Old Lycoptera Fossils Reveals Evolutionary Insights

8. Emerging Challenges in Molecular Paleontology: Misapplication of Environmental DNA Fragments and Misconception of Deamination as a Key Criterion for In Situ DNA Identification

9. ProtGO: A Transformer based Fusion Model for accurately predicting Gene Ontology (GO) Terms from full scale Protein Sequences

10. DART-Eval: A Comprehensive DNA Language Model Evaluation Benchmark on Regulatory DNA

q-bio.QM

1. Deep-learning-based identification of individual motion characteristics from upper-limb trajectories towards disorder stage evaluation

2. Decoding Drug Discovery: Exploring A-to-Z In silico Methods for Beginners

3. BarcodeMamba: State Space Models for Biodiversity Analysis

4. FlowDock: Geometric Flow Matching for Generative Protein-Ligand Docking and Affinity Prediction

5. Reliable and superior elliptic Fourier descriptor normalization and its application software ElliShape with efficient image processing

6. MEATRD: Multimodal Anomalous Tissue Region Detection Enhanced with Spatial Transcriptomics

7. Cardiovascular Disease Detection By Leveraging Semi-Supervised Learning

8. Predictive Pattern Recognition Techniques Towards Spatiotemporal Representation of Plant Growth in Simulated and Controlled Environments: A Comprehensive Review

9. RAID-Database: human Responses to Affine Image Distortions

10. MiCull2 – simulating mastitis transmission through milking order

q-bio.PE

1. Asymmetric Interactions Shape Survival During Population Range Expansions

2. Quasispecies dynamics with time lags and periodic fluctuations in replication

3. Explicit modeling of density dependence in spatial capture-recapture models

4. Stochastic models in phylogenetic comparative methods: analytical properties and parameter estimation

5. Multivariate Aspects of Phylogenetic Comparative Methods

6. The expensive son hypothesis

7. Self-similarity in pandemic spread and fractal containment policies

8. Estimating excess mortality during the Covid-19 pandemic in Aotearoa New Zealand

9. An assessment of Alberta's strategy for controlling mountain pine beetle outbreaks

10. Mountain pine beetle struggles with jack pine: A mechanistic explanation for slowed range expansion in Alberta

q-fin.CP

1. S&P 500 Trend Prediction

2. Simulation of square-root processes made simple: applications to the Heston model

3. From Votes to Volatility Predicting the Stock Market on Election Day

4. SusGen-GPT: A Data-Centric LLM for Financial NLP and Sustainability Report Generation

5. FinGPT: Enhancing Sentiment-Based Stock Movement Prediction with Dissemination-Aware and Context-Enriched LLMs

6. Reciprocity in Interbank Markets

7. Integrative Analysis of Financial Market Sentiment Using CNN and GRU for Risk Prediction and Alert Systems

8. Financial Fine-tuning a Large Time Series Model

9. Geometric Deep Learning for Realized Covariance Matrix Forecasting

10. Isogeometric Analysis for the Pricing of Financial Derivatives with Nonlinear Models: Convertible Bonds and Options

q-fin.PM

1. Cost-aware Portfolios in a Large Universe of Assets

2. PolyModel for Hedge Funds' Portfolio Construction Using Machine Learning

3. Geometric Deep Learning for Realized Covariance Matrix Forecasting

4. LLMs for Time Series: an Application for Single Stocks and Statistical Arbitrage

5. A Joint Energy and Differentially-Private Smart Meter Data Market

6. Smart leverage? Rethinking the role of Leveraged Exchange Traded Funds in constructing portfolios to beat a benchmark

7. Correlation without Factors in Retail Cryptocurrency Markets

8. Turnover of investment portfolio via covariance matrix of returns

9. MILLION: A General Multi-Objective Framework with Controllable Risk for Portfolio Management

10. Dynamic ETF Portfolio Optimization Using enhanced Transformer-Based Models for Covariance and Semi-Covariance Prediction(Work in Progress)

q-fin.TR

1. Auto-Regressive Control of Execution Costs

2. FinGPT: Enhancing Sentiment-Based Stock Movement Prediction with Dissemination-Aware and Context-Enriched LLMs

3. Efficient and Verified Continuous Double Auctions

4. A Joint Energy and Differentially-Private Smart Meter Data Market

5. A theory of passive market impact

6. Uncertain Regulations, Definite Impacts: The Impact of the US Securities and Exchange Commission's Regulatory Interventions on Crypto Assets

7. Ergodic optimal liquidations in DeFi

8. MarketGPT: Developing a Pre-trained transformer (GPT) for Modeling Financial Time Series

9. Calculating Profits and Losses for Algorithmic Trading Strategies: A Short Guide

10. Market Making without Regret

stat.AP

1. But Can You Use It? Design Recommendations for Differentially Private Interactive Systems

2. Efficient Bayesian inversion for simultaneous estimation of geometry and spatial field using the Karhunen-Loève expansion

3. Chopin: An Open Source R-language Tool to Support Spatial Analysis on Parallelizable Infrastructure

4. Spatial Cross-Recurrence Quantification Analysis for Multi-Platform Contact Tracing and Epidemiology Research

5. P3LS: Point Process Partial Least Squares

6. Missing data imputation for noisy time-series data and applications in healthcare

7. Balancing Accuracy and Costs in Cross-Temporal Hierarchies: Investigating Decision-Based and Validation-Based Reconciliation

8. Statistical Problems in the Diagnosis of Shaken Baby Syndrome/Abusive Head Trauma: Limitations to Algorithms and the Need for Reliable Data

9. CESAR: A Convolutional Echo State AutoencodeR for High-Resolution Wind Forecasting

10. Cardiovascular Disease Detection By Leveraging Semi-Supervised Learning

stat.ML

1. Generalization Analysis for Deep Contrastive Representation Learning

2. Multiplex Dirichlet stochastic block model for clustering multidimensional compositional networks

3. BetaExplainer: A Probabilistic Method to Explain Graph Neural Networks

4. Bayesian Surrogate Training on Multiple Data Sources: A Hybrid Modeling Strategy

5. Scalable Temporal Anomaly Causality Discovery in Large Systems: Achieving Computational Efficiency with Binary Anomaly Flag Data

6. Conditional Diffusion Models Based Conditional Independence Testing

7. Generalized Bayesian deep reinforcement learning

8. A partial likelihood approach to tree-based density modeling and its application in Bayesian inference

9. A Mapper Algorithm with implicit intervals and its optimization

10. Learning Massive-scale Partial Correlation Networks in Clinical Multi-omics Studies with HP-ACCORD

stat.TH

1. Optimality of the Right-Invariant Prior

2. The entropic optimal (self-)transport problem: Limit distributions for decreasing regularization with application to score function estimation

3. Causal Invariance Learning via Efficient Optimization of a Nonconvex Objective

4. A partial likelihood approach to tree-based density modeling and its application in Bayesian inference

5. Dual Unscented Kalman Filter Architecture for Sensor Fusion in Water Networks Leak Localization

6. Learning Massive-scale Partial Correlation Networks in Clinical Multi-omics Studies with HP-ACCORD

7. Well-Posedness and Stability of the Stochastic OGTT Model

8. Posterior asymptotics of high-dimensional spiked covariance model with inverse-Wishart prior

9. Model checking for high dimensional generalized linear models based on random projections

10. The Stein-log-Sobolev inequality and the exponential rate of convergence for the continuous Stein variational gradient descent method

eess.IV

1. Are the Latent Representations of Foundation Models for Pathology Invariant to Rotation?

2. Towards Physically-Based Sky-Modeling

3. Ant Nest Detection Using Underground P-Band TomoSAR

4. Ensemble Learning and 3D Pix2Pix for Comprehensive Brain Tumor Analysis in Multimodal MRI

5. Point Cloud-Assisted Neural Image Compression

6. Flex-PE: Flexible and SIMD Multi-Precision Processing Element for AI Workloads

7. Fast-staged CNN Model for Accurate pulmonary diseases and Lung cancer detection

8. High-speed and High-quality Vision Reconstruction of Spike Camera with Spike Stability Theorem

9. Data-driven Precipitation Nowcasting Using Satellite Imagery

10. Block-Based Multi-Scale Image Rescaling

eess.SP

1. Rate-Splitting Multiple Access for Integrated Sensing and Communications: A First Experimental Study

2. Soil moisture estimation of bare and vegetation-covered areas using a P/L/C-band SAR

3. Ant Nest Detection Using Underground P-Band TomoSAR

4. Scalable Data Transmission Framework for Earth Observation Satellites with Channel Adaptation

5. Sonar-based Deep Learning in Underwater Robotics: Overview, Robustness and Challenges

6. Evaluating the Efficacy of Vectocardiographic and ECG Parameters for Efficient Tertiary Cardiology Care Allocation Using Decision Tree Analysis

7. Acceleration and Parallelization Methods for ISRS EGN Model

8. On-the-Fly Interrogation of Mobile Passive Sensors from the Fusion of Optical and Radar Data

9. Capacity Analysis on OAM-Based Wireless Communications: An Electromagnetic Information Theory Perspective

10. Probabilistic GOSPA: A Metric for Performance Evaluation of Multi-Object Filters with Uncertainties

econ.EM

1. Moderating the Mediation Bootstrap for Causal Inference

2. VAR models with an index structure: A survey with new results

2026 Under review as a conference paper at ICLR 2026

3. Treatment Evaluation at the Intensive and Extensive Margins

4. Forecasting realized covariances using HAR-type models

5. Do LLMs Act as Repositories of Causal Knowledge?

6. An overview of meta-analytic methods for economic research

7. A Neyman-Orthogonalization Approach to the Incidental Parameter Problem

8. Geometric Deep Learning for Realized Covariance Matrix Forecasting

9. A Kernel Score Perspective on Forecast Disagreement and the Linear Pool

10. The Global Carbon Budget as a cointegrated system

econ.GN

1. Multiplexing in Networks and Diffusion

2. Transition dynamics of electricity asset-owning firms

3. Binary or nonbinary? An evolutionary learning approach to gender identity

4. On Prior Confidence and Belief Updating

5. Strategically Acting on Information

6. Is Polarization an Inevitable Outcome of Similarity-Based Content Recommendations? – Mathematical Proofs and Computational Validation

7. Re-examining the social impact of silver monetization in the Ming Dynasty from the perspective of supply and demand

8. Delving into Youth Perspectives on In-game Gambling-like Elements: A Proof-of-Concept Study Utilising Large Language Models for Analysing User-Generated Text Data

9. Does Low Spoilage Under Cold Conditions Foster Cultural Complexity During the Foraging Era? – A Theoretical and Computational Inquiry

10. Emulating the Global Change Analysis Model with Deep Learning

Table 1: Performance comparison for all-at-once on open-ended questions.

|      | K=1  | K=2  | K=3  | K=4  | K=5  | K=7  | K=10 |
|------|------|------|------|------|------|------|------|
| Q0:  | 0.39 | 0.4  | 0.4  | 0.4  | 0.41 | 0.42 | 0.56 |
| Q1:  | 0.63 | 0.66 | 0.66 | 0.65 | 0.63 | 0.64 | 0.57 |
| Q2:  | 0.41 | 0.43 | 0.41 | 0.4  | 0.36 | 0.41 | 0.41 |
| Q3:  | 0.5  | 0.49 | 0.6  | 0.67 | 0.67 | 0.66 | 0.78 |
| Q4:  | 0.33 | 0.46 | 0.4  | 0.43 | 0.38 | 0.42 | 0.42 |
| Q5:  | 0.39 | 0.41 | 0.47 | 0.3  | 0.31 | 0.42 | 0.43 |
| Q6:  | 0.45 | 0.78 | 0.78 | 0.79 | 0.78 | 0.78 | 0.75 |
| Q7:  | 0.84 | 0.84 | 0.84 | 0.84 | 0.84 | 0.83 | 0.83 |
| Q8:  | 0.39 | 0.41 | 0.47 | 0.36 | 0.37 | 0.37 | 0.39 |
| Q9:  | 0.47 | 0.46 | 0.43 | 0.46 | 0.48 | 0.45 | 0.44 |

## B.3 ADDITIONAL RESULTS FOR EXPLORE PUZZLES

We ran a robustness check on the first six open-ended question experiments using the universal sentence encoder of Cer (2018) as our embedding model. Our results remain largely unchanged, and are summarized in Figure 23.

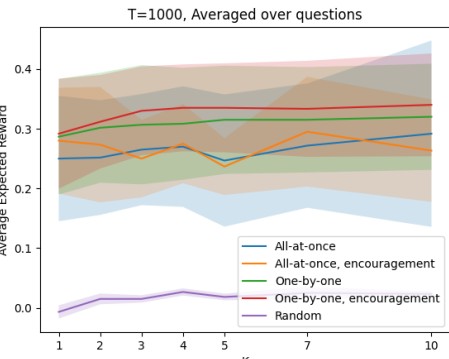

Figure 23: Results averaged over the first six questions, for embeddings generated using the universal sentence encoder.

Below are the individual plots for the remaining 9 questions using the Sentence-BERT encoder.

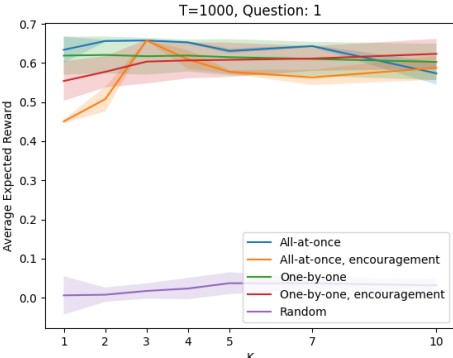

Figure 24: Results on Question 1

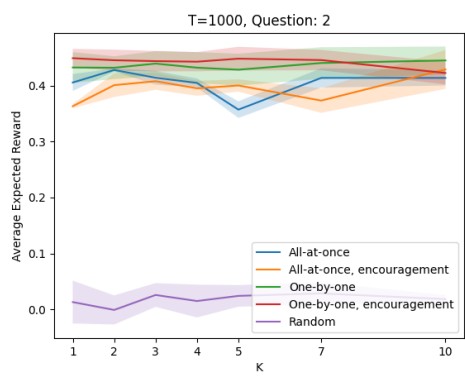

Figure 25: Results on Question 2

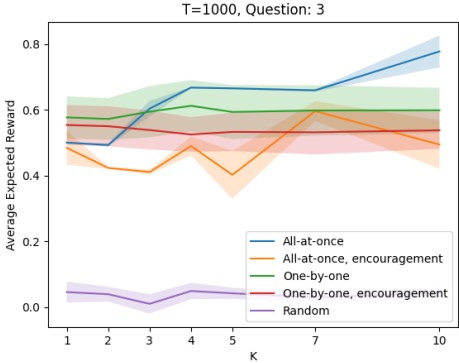

Figure 26: Results on Question 3

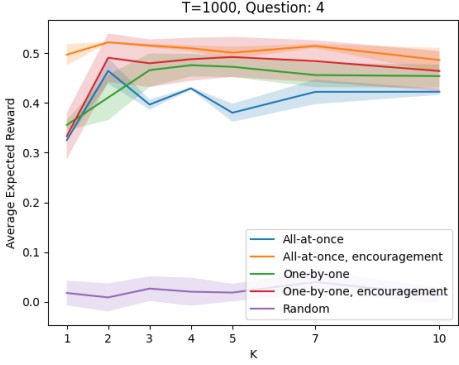

Figure 27: Results on Question 4

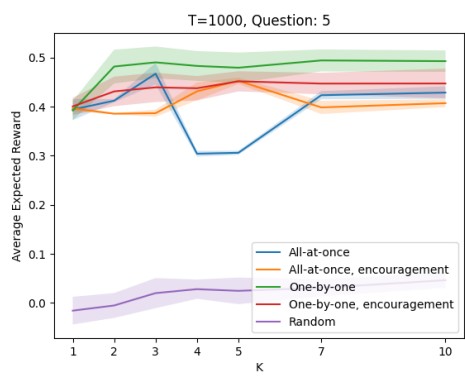

Figure 28: Results on Question 5

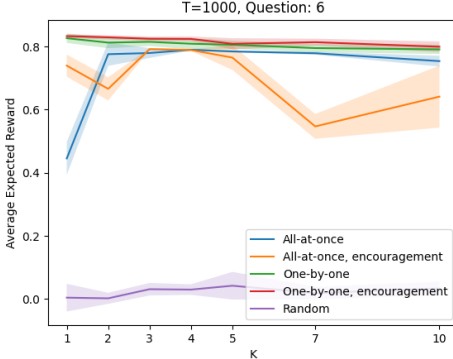

Figure 29: Results on Question 6

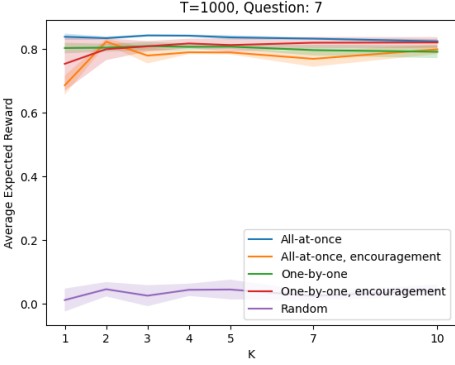

Figure 30: Results on Question 7

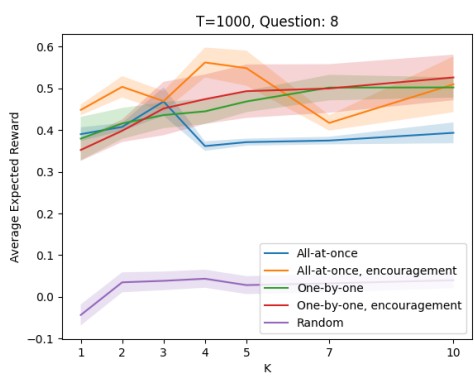

Figure 31: Results on Question 8

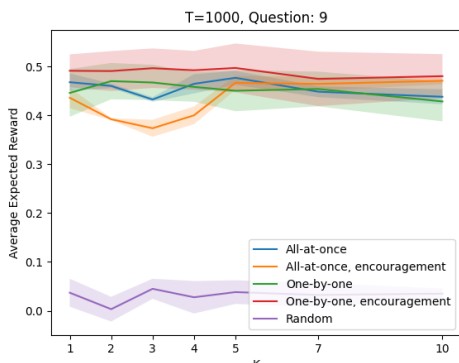

Figure 32: Results on Question 9

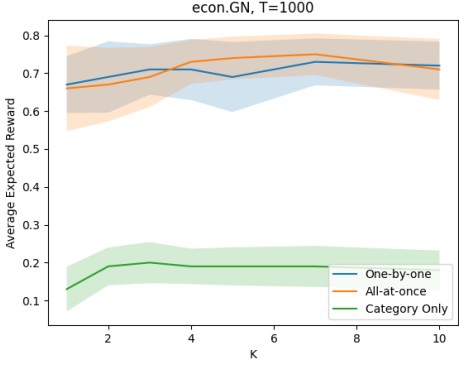

Figure 33: Results for arXiv category econ.GN

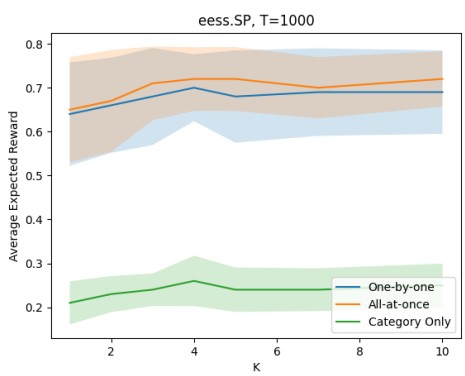

Figure 34: Results for arXiv category eess.SP

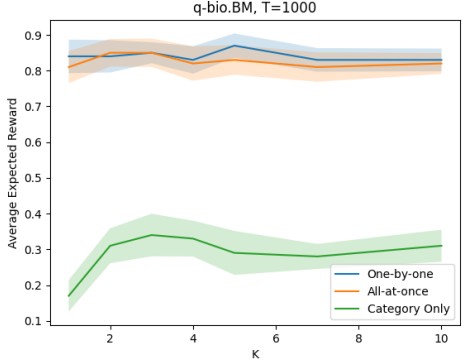

Figure 35: Results for arXiv category q-bio.BM

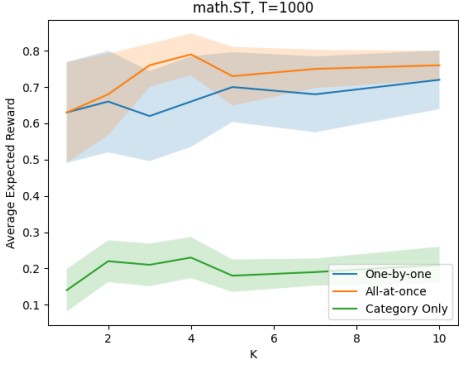

Figure 36: Results for arXiv category math.ST

Table 2: Performance comparison for all-at-once with encouragement on open-ended questions.

|     | K=1  | K=2  | K=3  | K=4  | K=5  | K=7  | K=10 |
|-----|------|------|------|------|------|------|------|
| Q0: | 0.4  | 0.4  | 0.34 | 0.35 | 0.37 | 0.52 | 0.55 |
| Q1: | 0.45 | 0.51 | 0.66 | 0.61 | 0.58 | 0.56 | 0.59 |
| Q2: | 0.36 | 0.4  | 0.41 | 0.4  | 0.4  | 0.37 | 0.43 |
| Q3: | 0.48 | 0.42 | 0.41 | 0.49 | 0.4  | 0.6  | 0.49 |
| Q4: | 0.5  | 0.52 | 0.52 | 0.51 | 0.5  | 0.51 | 0.49 |
| Q5: | 0.4  | 0.39 | 0.39 | 0.43 | 0.45 | 0.4  | 0.41 |
| Q6: | 0.74 | 0.67 | 0.79 | 0.79 | 0.77 | 0.55 | 0.64 |
| Q7: | 0.69 | 0.82 | 0.78 | 0.79 | 0.79 | 0.77 | 0.8  |
| Q8: | 0.45 | 0.5  | 0.47 | 0.56 | 0.55 | 0.42 | 0.51 |
| Q9: | 0.44 | 0.39 | 0.37 | 0.4  | 0.47 | 0.46 | 0.47 |

Table 3: Performance comparison for one-by-one on open-ended questions.

|     | K=1  | K=2  | K=3  | K=4  | K=5  | K=7  | K=10 |
|-----|------|------|------|------|------|------|------|
| Q0: | 0.33 | 0.4  | 0.4  | 0.43 | 0.43 | 0.47 | 0.52 |
| Q1: | 0.62 | 0.62 | 0.62 | 0.62 | 0.61 | 0.61 | 0.6  |
| Q2: | 0.43 | 0.43 | 0.44 | 0.43 | 0.43 | 0.44 | 0.45 |
| Q3: | 0.58 | 0.57 | 0.59 | 0.61 | 0.59 | 0.6  | 0.6  |
| Q4: | 0.36 | 0.41 | 0.47 | 0.48 | 0.47 | 0.46 | 0.45 |
| Q5: | 0.39 | 0.48 | 0.49 | 0.48 | 0.48 | 0.49 | 0.49 |
| Q6: | 0.83 | 0.81 | 0.82 | 0.81 | 0.81 | 0.8  | 0.79 |
| Q7: | 0.8  | 0.81 | 0.81 | 0.81 | 0.81 | 0.8  | 0.79 |
| Q8: | 0.38 | 0.42 | 0.44 | 0.44 | 0.47 | 0.5  | 0.5  |
| Q9: | 0.45 | 0.47 | 0.47 | 0.46 | 0.45 | 0.45 | 0.43 |

Table 4: Performance comparison for one-by-one with encouragement on open-ended questions.

|     | K=1  | K=2  | K=3  | K=4  | K=5  | K=7  | K=10 |
|-----|------|------|------|------|------|------|------|
| Q0: | 0.35 | 0.37 | 0.5  | 0.54 | 0.55 | 0.58 | 0.59 |
| Q1: | 0.55 | 0.58 | 0.6  | 0.61 | 0.61 | 0.61 | 0.62 |
| Q2: | 0.45 | 0.45 | 0.44 | 0.44 | 0.45 | 0.45 | 0.42 |
| Q3: | 0.55 | 0.55 | 0.54 | 0.53 | 0.53 | 0.53 | 0.54 |
| Q4: | 0.33 | 0.49 | 0.48 | 0.49 | 0.49 | 0.48 | 0.46 |
| Q5: | 0.4  | 0.43 | 0.44 | 0.44 | 0.45 | 0.45 | 0.45 |
| Q6: | 0.83 | 0.83 | 0.82 | 0.82 | 0.81 | 0.81 | 0.8  |
| Q7: | 0.75 | 0.8  | 0.81 | 0.82 | 0.81 | 0.82 | 0.82 |
| Q8: | 0.35 | 0.4  | 0.45 | 0.47 | 0.49 | 0.5  | 0.53 |
| Q9: | 0.49 | 0.49 | 0.5  | 0.49 | 0.5  | 0.47 | 0.48 |

Table 5: Performance comparison for random actions on open-ended questions.

|     | K=1   | K=2   | K=3  | K=4  | K=5  | K=7  | K=10 |
|-----|-------|-------|------|------|------|------|------|
| Q0: | -0.01 | -0.0  | 0.01 | 0.01 | 0.03 | 0.03 | 0.02 |
| Q1: | 0.01  | 0.01  | 0.02 | 0.02 | 0.04 | 0.04 | 0.03 |
| Q2: | 0.01  | -0.0  | 0.03 | 0.02 | 0.02 | 0.03 | 0.02 |
| Q3: | 0.05  | 0.04  | 0.01 | 0.05 | 0.04 | 0.03 | 0.04 |
| Q4: | 0.02  | 0.01  | 0.03 | 0.02 | 0.02 | 0.04 | 0.01 |
| Q5: | -0.02 | -0.01 | 0.02 | 0.03 | 0.02 | 0.03 | 0.05 |
| Q6: | 0.0   | 0.0   | 0.03 | 0.03 | 0.04 | 0.02 | 0.03 |
| Q7: | 0.01  | 0.05  | 0.03 | 0.04 | 0.04 | 0.02 | 0.04 |
| Q8: | -0.04 | 0.03  | 0.04 | 0.04 | 0.03 | 0.03 | 0.04 |
| Q9: | 0.04  | 0.0   | 0.04 | 0.03 | 0.04 | 0.03 | 0.04 |

Table 6: Performance for all-at-once on arXiv tasks.

|         | K=1  | K=2  | K=5  |
|---------|------|------|------|
| gr-qc   | 0.63 | 0.64 | 0.68 |
| hep-ex  | 0.81 | 0.81 | 0.83 |
| hep-lat | 0.72 | 0.72 | 0.72 |
| hep-ph  | 0.7  | 0.76 | 0.75 |
| hep-th  | 0.65 | 0.71 | 0.73 |
| math-ph | 0.64 | 0.73 | 0.74 |
| nucl-ex | 0.73 | 0.79 | 0.75 |
| nucl-th | 0.65 | 0.69 | 0.71 |
| quant-ph| 0.68 | 0.71 | 0.75 |
| cs.AI   | 0.66 | 0.71 | 0.72 |
| cs.CL   | 0.66 | 0.71 | 0.75 |
| cs.CV   | 0.72 | 0.74 | 0.71 |
| cs.LG   | 0.68 | 0.72 | 0.74 |
| cs.NE   | 0.71 | 0.78 | 0.78 |
| cs.RO   | 0.76 | 0.79 | 0.78 |
| cs.IT   | 0.72 | 0.73 | 0.72 |
| cs.CR   | 0.7  | 0.72 | 0.74 |
| cs.DS   | 0.75 | 0.77 | 0.77 |
| cs.HC   | 0.75 | 0.75 | 0.75 |
| math.AG | 0.7  | 0.78 | 0.78 |
| math.AT | 0.68 | 0.7  | 0.71 |
| math.AP | 0.7  | 0.79 | 0.78 |
| math.CT | 0.65 | 0.69 | 0.71 |
| math.GR | 0.73 | 0.77 | 0.76 |
| math.NT | 0.73 | 0.79 | 0.77 |
| math.OC | 0.77 | 0.79 | 0.76 |
| math.ST | 0.63 | 0.68 | 0.73 |
| q-bio.BM| 0.81 | 0.85 | 0.83 |
| q-bio.GN| 0.76 | 0.78 | 0.79 |
| q-bio.QM| 0.76 | 0.78 | 0.78 |
| q-bio.PE| 0.8  | 0.82 | 0.8  |
| q-fin.CP| 0.74 | 0.78 | 0.77 |
| q-fin.PM| 0.74 | 0.77 | 0.78 |
| q-fin.TR| 0.74 | 0.78 | 0.78 |
| stat.AP | 0.73 | 0.69 | 0.75 |
| stat.ML | 0.7  | 0.73 | 0.74 |
| stat.TH | 0.65 | 0.67 | 0.78 |
| eess.IV | 0.67 | 0.73 | 0.72 |
| eess.SP | 0.65 | 0.67 | 0.72 |
| econ.EM | 0.62 | 0.68 | 0.7  |
| econ.GN | 0.66 | 0.67 | 0.74 |

Table 7: Performance for one-by-one on arXiv tasks.

|          | K=1  | K=2  | K=5  |
|----------|------|------|------|
| gr-qc    | 0.63 | 0.64 | 0.65 |
| hep-ex   | 0.78 | 0.76 | 0.81 |
| hep-lat  | 0.72 | 0.72 | 0.74 |
| hep-ph   | 0.7  | 0.72 | 0.73 |
| hep-th   | 0.64 | 0.69 | 0.68 |
| math-ph  | 0.65 | 0.71 | 0.71 |
| nucl-ex  | 0.72 | 0.74 | 0.76 |
| nucl-th  | 0.64 | 0.67 | 0.71 |
| quant-ph | 0.7  | 0.71 | 0.71 |
| cs.AI    | 0.7  | 0.74 | 0.73 |
| cs.CL    | 0.69 | 0.71 | 0.75 |
| cs.CV    | 0.73 | 0.74 | 0.77 |
| cs.LG    | 0.67 | 0.72 | 0.72 |
| cs.NE    | 0.74 | 0.76 | 0.77 |
| cs.RO    | 0.78 | 0.79 | 0.78 |
| cs.IT    | 0.76 | 0.75 | 0.75 |
| cs.CR    | 0.72 | 0.71 | 0.74 |
| cs.DS    | 0.75 | 0.78 | 0.78 |
| cs.HC    | 0.72 | 0.72 | 0.73 |
| math.AG  | 0.68 | 0.77 | 0.78 |
| math.AT  | 0.63 | 0.69 | 0.7  |
| math.AP  | 0.72 | 0.75 | 0.76 |
| math.CT  | 0.62 | 0.71 | 0.73 |
| math.GR  | 0.71 | 0.76 | 0.74 |
| math.NT  | 0.73 | 0.75 | 0.73 |
| math.OC  | 0.73 | 0.77 | 0.8  |
| math.ST  | 0.63 | 0.66 | 0.7  |
| q-bio.BM | 0.84 | 0.84 | 0.87 |
| q-bio.GN | 0.69 | 0.76 | 0.75 |
| q-bio.QM | 0.76 | 0.8  | 0.77 |
| q-bio.PE | 0.8  | 0.79 | 0.82 |
| q-fin.CP | 0.71 | 0.72 | 0.76 |
| q-fin.PM | 0.67 | 0.77 | 0.73 |
| q-fin.TR | 0.72 | 0.74 | 0.75 |
| stat.AP  | 0.73 | 0.72 | 0.79 |
| stat.ML  | 0.74 | 0.75 | 0.77 |
| stat.TH  | 0.61 | 0.64 | 0.72 |
| eess.IV  | 0.71 | 0.72 | 0.74 |
| eess.SP  | 0.64 | 0.66 | 0.68 |
| econ.EM  | 0.66 | 0.66 | 0.67 |
| econ.GN  | 0.67 | 0.69 | 0.69 |

Table 8: Performance for Category Only baseline on arXiv tasks.

|  | K=1 | K=2 | K=5 |
|---|---|---|---|
| gr-qc | 0.28 | 0.32 | 0.36 |
| hep-ex | 0.25 | 0.35 | 0.38 |
| hep-lat | 0.31 | 0.31 | 0.38 |
| hep-ph | 0.25 | 0.26 | 0.32 |
| hep-th | 0.24 | 0.26 | 0.27 |
| math-ph | 0.22 | 0.3 | 0.3 |
| nucl-ex | 0.37 | 0.38 | 0.35 |
| nucl-th | 0.31 | 0.32 | 0.33 |
| quant-ph | 0.23 | 0.27 | 0.27 |
| cs.AI | 0.15 | 0.17 | 0.18 |
| cs.CL | 0.14 | 0.19 | 0.26 |
| cs.CV | 0.19 | 0.22 | 0.32 |
| cs.LG | 0.21 | 0.19 | 0.24 |
| cs.NE | 0.27 | 0.31 | 0.3 |
| cs.RO | 0.27 | 0.27 | 0.31 |
| cs.IT | 0.29 | 0.31 | 0.34 |
| cs.CR | 0.21 | 0.24 | 0.32 |
| cs.DS | 0.2 | 0.2 | 0.22 |
| cs.HC | 0.12 | 0.2 | 0.23 |
| math.AG | 0.32 | 0.33 | 0.33 |
| math.AT | 0.33 | 0.34 | 0.4 |
| math.AP | 0.19 | 0.23 | 0.32 |
| math.CT | 0.25 | 0.23 | 0.29 |
| math.GR | 0.25 | 0.28 | 0.34 |
| math.NT | 0.21 | 0.29 | 0.32 |
| math.OC | 0.2 | 0.27 | 0.3 |
| math.ST | 0.14 | 0.22 | 0.18 |
| q-bio.BM | 0.17 | 0.31 | 0.29 |
| q-bio.GN | 0.2 | 0.25 | 0.32 |
| q-bio.QM | 0.03 | 0.12 | 0.1 |
| q-bio.PE | 0.28 | 0.32 | 0.31 |
| q-fin.CP | 0.34 | 0.36 | 0.37 |
| q-fin.PM | 0.39 | 0.39 | 0.43 |
| q-fin.TR | 0.33 | 0.34 | 0.38 |
| stat.AP | 0.07 | 0.12 | 0.1 |
| stat.ML | 0.19 | 0.21 | 0.28 |
| stat.TH | -0.01 | 0.07 | 0.12 |
| eess.IV | 0.18 | 0.19 | 0.24 |
| eess.SP | 0.21 | 0.23 | 0.24 |
| econ.EM | 0.21 | 0.32 | 0.37 |
| econ.GN | 0.13 | 0.19 | 0.19 |

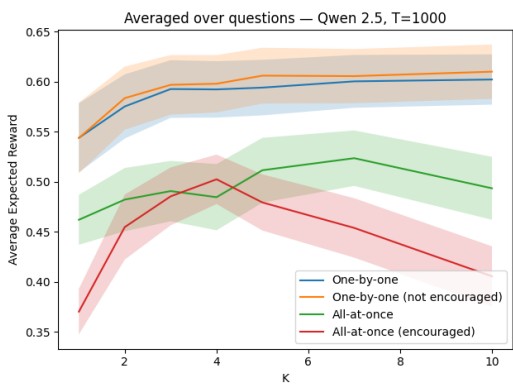

Figure 37: Results for QWEN-2.5 in the setting of Figure 6.

## B.4 BENCHMARKING ENCODERS

Here we benchmark the two encoders we use (Sentence-BERT and the universal sentence encoder) by measuring the cosine similarity between semantically similar/different words.

Table 9: Cosine similarity of different words.

|  | Sentence-BERT | Universal Sentence Encoder |
|---|---|---|
| dog, tacos: | 0.25 | 0.24 |
| Pittsburgh, tiki bar: | 0.12 | 0.17 |
| Honolulu, tiki bar: | 0.30 | 0.25 |
| Pittsburgh, Honolulu: | 0.41 | 0.29 |
| angel, devil: | 0.48 | 0.54 |
| machine learning, artificial intelligence: | 0.70 | 0.58 |
| war, peace: | 0.61 | 0.49 |
| love, hate: | 0.49 | 0.59 |
| love, affection: | 0.62 | 0.56 |
| war, battle: | 0.74 | 0.57 |
| machine learning, battle: | 0.25 | 0.19 |

The similarity scores of both models in Table 9 suggest that while the embeddings produced by both embedding models are generally "in the ballpark" of what one would consider "similar"/"different", they are still a somewhat coarse measure of distance, which may explain the similar performance of our different prompting strategies.

## B.5 ROBUSTNESS CHECK WITH OTHER MODELS

In this appendix, we include results for Qwen2.5-7B-Instruct (QWEN-2.5), Gemma 3 12B (GEMMA-3), and Mistral-7B-Instruct-v0.3 (MISTRAL-7B) on our explore Q/A puzzles. As was the case with the GPT models, performance increased with $K$ when the candidate answers are generated one-by-one. When generating answers all-at-once, we observed that QWEN-2.5 and MISTRAL-7B had trouble generating concise answers, leading to a drop in performance as $K$ increases.

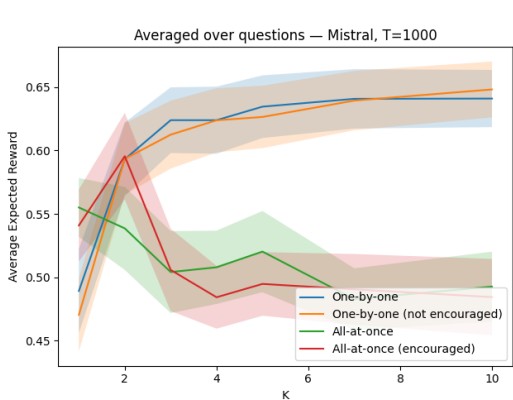

Figure 38: Results for MISTRAL-7B in the setting of Figure 6.

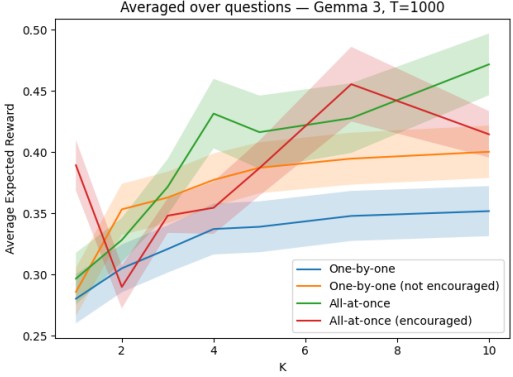

Figure 39: Results for GEMMA-3 in the setting of Figure 6.

### B.6 EXPLORE EXPERIMENTS ON THE MOVIE LENS DATASET

**Experiment setup:** We used the Movie Lens 100K dataset Harper & Konstan (2015), which contains ratings from 943 users across 1682 movies, to model a stylized movie recommendation task. Given a user, we run a multi-armed bandit algorithm over $T$ rounds, where arms are movies and rewards are random Bernoulli draws parameterized by the users' ground-truth ratings. Since in the dataset every user did not rate every movie, we used a standard algorithm to complete the data matrix, namely the soft-impute low-rank matrix completion method from the fancyimpute Python package. We then normalized this matrix so that all values were between 0 and 1.

Next we prompted the LLM to discretize the set of movies, i.e. to generate a representative subset of size $K \in \{10, 19\}$. For each of the 943 users, we then ran a multi-armed bandit algorithm (UCB1) over these K recommendations for $T = 1000$ rounds, and averaged the cumulative rewards over all users. For baselines, we compared against running UCB1 over (1) a random subset of K recommendations (different random draws for each user), and (2) a random movie from each of the 19 movie genres in the dataset (also different random draws for each user). For each algorithm and baseline, we repeated the MAB procedure 5 times and averaged over time, to de-noise.

Our results for MISTRAL-7B and QWEN-2.5 are given in Table 10 (all numbers are average cumulative rewards). We also tried running the experiment using LLAMA-3-8B and GEMMA-3, but the list of all movies was too long to fit in the context window.

| Model | Avg Rew ($K = 10$) | Avg Rew ($K = 19$) |
|---|---|---|
| Qwen | 671.88 | 678.25 |
| Mistral | 735.38 | 675.16 |
| Random Selection | 570.39 | 599.73 |
| Genre-based | — | 617.27 |

Table 10: Comparison of results for $K = 10$ and $K = 19$

Thus, both LLMs significantly outperform all baselines for both values of $K$.

Unlike in our other explore experiments, the LLM has implicit knowledge about each of the movies in the database, which it may leverage when selecting the discretization. In order to mitigate the chance that the LLM overfits to "popular" movies, we included instructions in the prompt explicitly stating to not take movie popularity into account. However, we didn't notice a significant difference in rewards from doing this versus not.

Prompt:

> [SYSTEM] You are a movie expert helping a user choose a movie.
>
> [USER] Here is a list of movies with their numeric IDs:{movie_list_str}
>
> From this list, choose K movies. You don't know what taste in movies the user has, so select a diverse set of movies from different genres such that they will most likely enjoy at least one of the movies you select. Respond ONLY with the K numeric IDs, one per line, with no extra text. Do NOT consider a movie's popularity when deciding whether to select it.

