# OpenReview forum: "Should You Use Your LLM to Explore or Exploit?"
_ICLR.cc/2026/Conference — Submitted to ICLR 2026_

### Official Review · Reviewer_YRTL · 2025-10-16

**Soundness:** 2
**Presentation:** 3
**Contribution:** 1
**Rating:** 2
**Confidence:** 4

**Summary:**

This paper studies whether LLMs should be used for exploitation or exploration in bandit-style decision problems. In this  paper, "exploitation" means let LLMs directly pick the best action from histories, and  "exploration" means let LLMs propose actions worth trying. In multi-armed bandits and contextual bandits, LLMs consistently underperform linear regression at exploitation, even with COT and natural mitigations (history compression via k-nearest/k-means). For exploration in large textual action spaces, treating LLM as a candidate arm generator is effective. The paper's contribution is that it shows LLMs are weak at exploitation but can be useful at proposing candidate arms in textual action spaces.

**Strengths:**

1. This paper cleanly separates exploitation vs. exploration roles for LLMs in bandit-style decision making, then study separately. This framing is clear.
2. This paper shows in-context history compression strategies are effective in the scenario of using LLMs for exploitation, which is a meaningful guidance.
3. The results give a concrete role assignment for LLMs in decision systems: avoid using them as exploitation oracles, but leverage them as candidate generators in large textual action spaces.
4. Clear task constructions and prompt templates make the work easy to understand.

**Weaknesses:**

1. In brief, the conclusions are predictable and not useful. The two central takeaways (LLMs underperform simple regression for exploitation, and LLMs can generate reasonable textual candidates) are easy to anticipate. The former favors stable numeric estimation where classic methods excel; the latter is essentially semantic discretization where LLMs have strong language priors, so beating crude controls is unsurprising.
2. Missing online evaluation of the exploration–exploitation trade-off. Bandits are about multi-round online balancing of exploration and exploitation and minimizing cumulative regret. This paper measures two isolated, one-shot oracles (pick an arm once; generate a candidate set once), leaving the key question unanswered: if an LLM actually participates in the decision loop—as a prior/uncertainty provider or an adaptive proposer—does it reduce regret, improve sample efficiency, and remain stable under fixed budgets? To answer this, this paper should evaluate LLM-guided bandit algorithms in sequence-decision scenarios, and report regret and token/latency budgets. Krishnamurthy et al. [1] have done some valuable work in this area, but this paper did not delve deeper into the exploration exploitation trade-off capability of LLMs.
3. Overall, the paper’s framework (testing LLM exploitation and exploration in isolation) makes the experiments largely uninformative for bandits and yields near-zero contribution. First, in the MAB setting, the finding (“performance degrades (1) as history length T increases and (2) as the empirical gap decreases”) is not useful: for (1), modern LLMs can explicitly execute code and compute means/variances, so scaling to long histories is easy; for (2), of course the probability of picking the best arm drops as the task gets harder—this needs no experiment. Second, in the CB setting, this paper effectively shows that LLM supervised regression underperforms linear regression, which is unsurprising and misaligned with current practice where LLMs can readily call code/external tools; moreover, the paper offers no remedy for scaling d and K. Third, the LLMs as exploration oracles experiments merely demonstrate that LLMs can propose a few plausible text candidates as arms—an almost self-evident claim that hardly requires experimentation.
4. Exploration baselines are too weak. The strongest comparator is “random candidates”, implemented by uniformly sampling K vectors in the sentence-embedding space and treating them as arms, with cosine similarity to a target vector as reward. This baseline is not even natural-language; the candidates are typically semantically meaningless and thus extremely weak. LLM has rich prior knowledge, so experiments using such a baseline cannot explain anything.

[1] Akshay Krishnamurthy, Keegan Harris, Dylan J. Foster, Cyril Zhang, and Aleksandrs Slivkins. Can
large language models explore in-context? In NeurIPS, 2024.

**Questions:**

1. Can you explain why do you ignore the exploration–exploitation trade-off and study the ability of LLMs to explore and exploit in-context in silos?
2. Your random baseline samples vectors uniformly in embedding space (not on the natural-language manifold). Can you add some reasonable baselines?

---

> ### Author Response · Authors · 2025-11-18
>
> Thanks for taking the time to review our submission. Please find our replies below.
>
> > *In brief, the conclusions are predictable and not useful. The two central takeaways (LLMs underperform simple regression for exploitation, and LLMs can generate reasonable textual candidates) are easy to anticipate. The former favors stable numeric estimation where classic methods excel; the latter is essentially semantic discretization where LLMs have strong language priors, so beating crude controls is unsurprising.*
>
> We disagree with your characterization of our results. It was not a priori clear to us that LLMs would be out-performed by a simple linear regression on exploitation tasks, especially when the underlying reward-generating function is non-linear and when the LLMs are aided by our in-context mitigations. Moreover, our exploitation experiments were designed precisely to characterize the boundary of tractability for current models. Figures 1–5 show that failures emerge even for relatively short histories and relatively large gaps, which is not something one could have confidently predicted in advance. The only hard evidence we had previously came from sequential MAB tasks, where an LLM was the top decision-maker and potential exploitation and exploration failures were conflated.
> Regarding exploration, while it may seem intuitive in hindsight that LLMs should outperform crude baselines when the action space has rich semantic structure, our results quantify the extent of that advantage, show how it propagates into performance on downstream bandit tasks (Figures 6-8), and show that this behavior is robust across different prompting strategies. We would also like to point out that it is not obvious what the relevant baseline to compare against should be, beyond a uniform discretization of the action space.
>
> In fact, uniformly sampling in the embedding space is arguably an unusually strong baseline, in some sense. The embedding already reflects the SOTA semantics. Precisely because the baseline does not operate in natural language, it enjoys an advantage that no LLM shares: it can perfectly and uniformly discretize the embedding space, ensuring a broad and unbiased coverage of the action space. LLM-generated candidates cannot do this, as they must remain in natural language and therefore inherit the semantic and structural biases of the model’s prior.
>
> > *Missing online evaluation of the exploration–exploitation trade-off. Bandits are about multi-round online balancing of exploration and exploitation and minimizing cumulative regret. This paper measures two isolated, one-shot oracles (pick an arm once; generate a candidate set once), leaving the key question unanswered: if an LLM actually participates in the decision loop—as a prior/uncertainty provider or an adaptive proposer—does it reduce regret, improve sample efficiency, and remain stable under fixed budgets? To answer this, this paper should evaluate LLM-guided bandit algorithms in sequence-decision scenarios, and report regret and token/latency budgets. Krishnamurthy et al. [1] have done some valuable work in this area, but this paper did not delve deeper into the exploration exploitation trade-off capability of LLMs.*
>
> While full online evaluation of an LLM decision-maker (DM) is an important research direction (which has been done previously), our goal in this paper is different. Instead, we focus on decomposing the classic exploration–exploitation problem into its two core computational primitives and evaluating them in isolation under controlled conditions. The purpose is to better understand the sucess/failure modes of the LLM DM and (more importantly) to study other alternative usage modes of LLMs (i.e., as  “oracles” rather than top-level DMs).
>
> Note that regret is a composite object: poor performance can arise from deficient exploration, deficient exploitation, or interactions between these components. So, it is not suitable metric for our “siloed” evaluation.

---

> ### Author Response · Authors · 2025-11-18
>
> > *Overall, the paper’s framework (testing LLM exploitation and exploration in isolation) makes the experiments largely uninformative for bandits and yields near-zero contribution. First, in the MAB setting, the finding (“performance degrades (1) as history length T increases and (2) as the empirical gap decreases”) is not useful: for (1), modern LLMs can explicitly execute code and compute means/variances, so scaling to long histories is easy; for (2), of course the probability of picking the best arm drops as the task gets harder—this needs no experiment. Second, in the CB setting, this paper effectively shows that LLM supervised regression underperforms linear regression, which is unsurprising and misaligned with current practice where LLMs can readily call code/external tools; moreover, the paper offers no remedy for scaling d and K. Third, the LLMs as exploration oracles experiments merely demonstrate that LLMs can propose a few plausible text candidates as arms—an almost self-evident claim that hardly requires experimentation.*
>
> The MAB puzzle is not presented as a standalone result but as a warm-up designed to establish a clean baseline before moving to the contextual bandit (CB) setting, which is the primary target of our analysis for exploitation. While it is indeed expected that failures occur for sufficiently large T, what was not known (and cannot be deduced from intuition alone) is where the success–failure boundary lies for modern LLMs, especially under in-context mitigations. Our experiments map out this boundary explicitly and show, for example, that failures arise even when both the history length and the empirical gap fall in seemingly tractable regimes.
> The statement “the probability of picking the best arm drops as the task gets harder” misses the point: the key question is not whether task difficulty affects performance but how quickly LLMs’ performance degrades and under what mitigations. In particular, we see sup-par LLM performance even for relatively large “gap”.
>
> In the CB setting, the critique that “this merely shows LLM regression underperforms linear regression” overlooks the motivation. If simple linear regression suffices, then one has no reason to invoke an LLM; the interesting question is whether LLMs add value when linear methods fail. That is why we included experiments with both linear and non-linear reward functions. In the non-linear setting (where linear regression struggles by construction) LLMs still underperform, despite receiving mitigations that condense and structure the history. This is a substantive, nontrivial finding about current model limitations. As for scaling d and K: our main takeaway is that modern LLMs do not currently scale in these dimensions. While our in-context mitigations are partial remedies, our primary contribution in this setting is to measure the limits of current models, and those limits are exactly what our results make visible.
>
> Finally, the exploration experiments do not merely show that LLMs can produce “a few plausible text candidates.” The difficulty is not in generating superficially relevant text but in generating candidate sets that adequately cover the action space with respect to a principled distance metric and that subsequently enable a downstream bandit algorithm (UCB1 in our case) to perform meaningfully better than an uninformed baseline.
>
> > *Exploration baselines are too weak. The strongest comparator is “random candidates”, implemented by uniformly sampling K vectors in the sentence-embedding space and treating them as arms, with cosine similarity to a target vector as reward. This baseline is not even natural-language; the candidates are typically semantically meaningless and thus extremely weak. LLM has rich prior knowledge, so experiments using such a baseline cannot explain anything.*
>
> Uniformly sampling in the embedding space is arguably an unusually strong baseline, in some sense. The embedding already reflects the SOTA semantics. Precisely because the baseline does not operate in natural language, it enjoys an advantage that no LLM shares: it can perfectly and uniformly discretize the embedding space, ensuring a broad and unbiased coverage of the action space. LLM-generated candidates cannot do this, as they must remain in natural language and therefore inherit the semantic and structural biases of the model’s prior.
>
> Moreover, it is not obvious what a stronger or more “natural” baseline would be. Designing a non-LLM text sampler that produces meaningful and diverse natural-language candidates at scale is itself a challenging problem, and any such heuristic would introduce domain-specific structure that would undercut the clean comparison we aim to make.

---

> ### Author Response · Authors · 2025-11-18
>
> > *Can you explain why do you ignore the exploration–exploitation trade-off and study the ability of LLMs to explore and exploit in-context in silos?*
>
> Please see our reply to your second stated weakness.
>
> > *Your random baseline samples vectors uniformly in embedding space (not on the natural-language manifold). Can you add some reasonable baselines?*
>
> Please see our reply to your fourth stated weakness.

---

> ### Comment · Reviewer_YRTL · 2025-11-21
> **I still think that your paper contribution is insufficient**
>
> Your paper has weak theoretical or practical value, and you have not answered my questions directly.
> Besides, it is obvious that the llm’s prior is stronger. And "the embedding already reflects the SOTA semantics" is obviously wrong.

---

> > ### Author Response · Authors · 2025-11-22
> >
> > If you have concrete suggestions for discretization baselines besides randomly sampling from the embedding space, we would be happy to compare against them.
> >
> > We attempted to answer your questions to the best of our ability, but if you believe there is an aspect of your review which we did not adequately address, we would be happy to elaborate.

---

> ### Comment · Reviewer_YRTL · 2025-11-27
>
> Thank you for the additional MovieLens experiment and for the detailed rebuttal. I also want to acknowledge that some of my earlier wording may have sounded stronger than intended; that was not my goal, and I appreciate your engagement.
>
> The new MovieLens results do help demonstrate that your exploration approach has some practical value in a more realistic setting. However, my main concerns remain: the motivation for evaluating exploitation and exploration in isolation is still not fully convincing, and the connection to actual online regret or full decision-loop performance remains unclear. The exploration baselines could also benefit from clearer justification, even if constructing stronger natural-language baselines is difficult.
>
> Overall, the clarifications and added experiment improve the paper, but I still feel the contribution could be better articulated and positioned. Thank you again for the thoughtful responses.

---

### Official Review · Reviewer_9szX · 2025-10-28

**Soundness:** 3
**Presentation:** 3
**Contribution:** 1
**Rating:** 4
**Confidence:** 3

**Summary:**

This paper evaluates the capacity of current large language models (LLMs) to aid decision-making agents facing the exploration-exploitation tradeoff. The study assesses these two functions independently within various (contextual) bandit frameworks. The findings reveal that LLMs struggle significantly with exploitation; even when in-context mitigations like k-nearest or k-means summarization are applied to small-scale tasks, LLM performance remains inferior to a simple linear regression baseline. In contrast, the research demonstrates that LLMs are highly effective at exploration, particularly in large action spaces with inherent semantic structure. In this capacity, LLMs can propose a concise set of high-quality candidate actions, and employing these candidates within an off-the-shelf bandit algorithm yields performance far superior to random baseline methods.

**Strengths:**

* The paper introduces a clear and practical framework for combining Large Language Models with bandit algorithms to address optimization problems in vast, semantically-rich action spaces. It effectively decouples the problem by leveraging the LLM as an "exploration oracle" to generate a high-quality, reduced set of candidate actions.   This set is then efficiently processed by a standard bandit algorithm (e.g., UCB1) for optimization. The feasibility and superiority of this approach are well-supported by thorough experimentation.

* The study is supported by a comprehensive suite of experiments that systematically evaluate LLM capabilities in sequential decision-making.

**Weaknesses:**

* The paper's novelty is severely undermined by its extensive reliance on the experimental framework established by Krishnamurthy et al. (2024) [1]. The core research question, the background framing, the "exploitation" task design (including the MAB puzzles), and even the prompting strategies appear to be heavily borrowed, if not directly replicated, from [1]. While this paper offers a different conclusion by separating exploration and exploitation "in silos," this distinction does not sufficiently justify re-using an almost identical research setup. This significant overlap in methodology and problem formulation makes the work appear derivative and substantially weakens its original contribution.

* Using LLMs as an "exploration oracle"—appears to be effective only within a limited domain. The success of this method hinges on the action space being inherently and richly semantic (e.g., free-text answers, paper titles), requiring a strong text-understanding capability from the LLM. This raises significant concerns about the generalizability of the approach. It is unclear whether this framework can be extended to other high-dimensional decision-making tasks that are not purely text-based or where the semantic structure is less explicit for an LLM to leverage effectively.

---
[1] Akshay Krishnamurthy, Keegan Harris, Dylan J. Foster, Cyril Zhang, and Aleksandrs Slivkins. Can large language models explore in-context? In *NeurIPS*, 2024.

**Questions:**

* The paper's conclusion that LLMs struggle with exploitation is a significant claim. Given the rapid improvements in the reasoning and numerical processing capabilities of new LLMs (e.g., Gemini-2.5-Flash, Gemini-2.5-Pro), it is possible that newer, more advanced models might exhibit fundamentally different performance on these tasks. Can you show whether this negative conclusion regarding exploitation remains valid for the absolute latest generation of models?

* The paper's primary positive finding relies on the LLM's role as an "exploration oracle" in tasks with inherently rich semantic action spaces (QA, arXiv titles). This raises a critical question about the method's broader applicability. Could this framework be adapted to more traditional, non-text-native bandit problems, such as classic recommender system benchmarks (e.g., the MovieLens dataset [1])? Specifically, if an LLM were used to propose a candidate set of items (e.g., movies) based on their metadata, could this approach  consistently outperform a pure contextual bandit algorithm?

---
[1] Harper, F. M. and Konstan, J. A. The movielens datasets: History and context. *Acm transactions on interactive intelligent systems (tiis)*, 5(4):1–19, 2015.

---

> ### Author Response · Authors · 2025-11-18
>
> Thanks for taking the time to review our work. Please find our responses to your questions and comments below.
>
> > *The paper's novelty is severely undermined by its extensive reliance on the experimental framework established by Krishnamurthy et al. (2024) [1]. The core research question, the background framing, the "exploitation" task design (including the MAB puzzles), and even the prompting strategies appear to be heavily borrowed, if not directly replicated, from [1]. While this paper offers a different conclusion by separating exploration and exploitation "in silos," this distinction does not sufficiently justify re-using an almost identical research setup. This significant overlap in methodology and problem formulation makes the work appear derivative and substantially weakens its original contribution.*
>
> While our experimental setup shares some surface-level similarities with Krishnamurthy et al. (2024), their findings and metrics only hold for small-arm MAB settings and do not extend to the more complex (contextual) bandit regimes we study.
> Furthermore, their work evaluates end-to-end agent performance, where multiple failure modes (e.g. reasoning errors, exploration mistakes) are entangled. Our contribution is to decompose these behaviors and evaluate exploitation and exploration separately under controlled conditions. This siloed evaluation framework is fundamentally different in purpose, usage mode, and scope: it enables us to diagnose why LLM agents fail or succeed in decision-making tasks, rather than merely observing that they do.
>
> > *Using LLMs as an "exploration oracle"—appears to be effective only within a limited domain. The success of this method hinges on the action space being inherently and richly semantic (e.g., free-text answers, paper titles), requiring a strong text-understanding capability from the LLM. This raises significant concerns about the generalizability of the approach. It is unclear whether this framework can be extended to other high-dimensional decision-making tasks that are not purely text-based or where the semantic structure is less explicit for an LLM to leverage effectively.*
>
> We do not view the reliance on semantic structure as a weakness. If an action space has no meaningful semantics for an LLM to exploit, then it is entirely reasonable that other approaches would be preferable. The value of our work is precisely that it targets the large class of decision-making problems where actions do have a rich linguistic or conceptual meaning. This is a setting in which LLMs demonstrably help (suggestively in prior work and more definitively in our experiments) and our results provide a clearer understanding of why and when that help materializes.
>
> > *The paper's conclusion that LLMs struggle with exploitation is a significant claim. Given the rapid improvements in the reasoning and numerical processing capabilities of new LLMs (e.g., Gemini-2.5-Flash, Gemini-2.5-Pro), it is possible that newer, more advanced models might exhibit fundamentally different performance on these tasks. Can you show whether this negative conclusion regarding exploitation remains valid for the absolute latest generation of models?*
>
> We agree that model capabilities are improving, and future systems may indeed behave differently. With that being said, the models we evaluate are already among the most capable at the time of submission, and we ask the reviewer to respect the practical realities of conference timelines. New models emerge continuously, and it is rarely feasible to rerun the full experimental suite each time a new model is released.
>
> More importantly, the contribution of this section is not to make a claim about every future LLM, but to show that models which are considered strong demonstrably fail at basic exploitation. This is a valuable finding in its own right: it cautions against assuming that in-context exploitation “just works”, and it highlights the need for explicit evaluation of this capability (whenever it is being relied upon for some new models) rather than taking it for granted.
>
> > *The paper's primary positive finding relies on the LLM's role as an "exploration oracle" in tasks with inherently rich semantic action spaces (QA, arXiv titles). This raises a critical question about the method's broader applicability. Could this framework be adapted to more traditional, non-text-native bandit problems, such as classic recommender system benchmarks (e.g., the MovieLens dataset [1])? Specifically, if an LLM were used to propose a candidate set of items (e.g., movies) based on their metadata, could this approach consistently outperform a pure contextual bandit algorithm?*
>
> We are happy to run more exploration experiments using the MovieLens dataset. We will post an update here in the following days.

---

### Official Review · Reviewer_dBmo · 2025-10-28

**Soundness:** 3
**Presentation:** 3
**Contribution:** 3
**Rating:** 6
**Confidence:** 4

**Summary:**

The paper offers behavioral and empirical results on evaluating frontier LLMs in decision-making under uncertainty. The authors isolate the abilities of LLMs to act as exploitation oracles (making the best decision given current data) and exploration oracles (trying new options for long-term benefit).

**Strengths:**

The paper has very clear theoretical framing and uses RL theory to guide empirical investigations -- this is applaudable.

I find the two testing situations interesting, especially the exploration case, where exploration in large action space has been very difficult for theory-driven algorithms and rely on aggressive assumptions to have efficient algorithms.

Despite many issues and questions I have raised, this paper is technically sound and useful for expanding the discourse on LLM for exploration/exploitation. Most of my issues are not difficult to fix during rebuttal and I hope to engage with the authors in a constructive way to enhance this paper for a broader audience.

**Weaknesses:**

**Lack of Real Domains**

The paper cited a few prior works -- in [1], they have expanded CB domain to include MovieLens, which is a common recommendation algorithm benchmark/dataset adapted for LLMs. The code has also been released.

It is understandable to do MAB experiments in synthetic/toy domain, but for CB, we should continue to incorporate realistic domains in healthcare, education, or recommendation systems. Understanding LLM's behavior in these domains is crucial, especially for a behavioral/empirical investigation like this.

The domain can be rather small, or the testing can be limited, but not incorporating them makes the paper less impactful and useful to the practitioners.

**Lack of New Metrics**

The paper investigated `FracCorrect` and `rew` as two metrics. `FracCorrect` is already investigated in an earlier paper [1] but was called `OptFrac`. Also, [2] proposed `MinFrac`, which is an interesting metric to understand exploration behavior. Do you guys have some thoughts about what other metrics are interesting to track model's exploration and exploitation behaviors? Is it easy to redo some analysis on already collected data to yield more insights?

**Misleading Names**

`DeepSeek-R1-Distill-Qwen-32B` should not be named as `DeepSeek-R1`. A 32B model cannot be compared to a 685B model...this might be too misleading. Please name it DeepSeek-R1-Distill. In my practical experience, these two models are very different.


[1] EVOLvE: Evaluating and Optimizing LLMs For Exploration
[2] Can Large Language Models Explore In-Context?

**Questions:**

1. Any chance to add a realistic domain for CB (especially for the exploration oracle)? For the exploitation case, I think prior work (and your current investigation) is sufficient to show LLMs don't work that well.
2. Prior works lack evaluation on thinking-models (DeepSeek-R1 -- full model, Qwen3, etc.) -- do you think you have time/ability to add some analysis on the full DeepSeek-R1, or re-use the data you saved from DeepSeek-R1-Distill-Qwen-32B and evaluating the quality of the chain of thoughts there (instead of doing CoT prompting, you can get the actual thinking tokens)?

---

> ### Author Response · Authors · 2025-11-18
>
> Thanks for taking the time to review our submission. Please find our replies to your questions and comments below.
>
> > *The paper investigated FracCorrect and rew as two metrics. FracCorrect is already investigated in an earlier paper [1] but was called OptFrac. Also, [2] proposed MinFrac, which is an interesting metric to understand exploration behavior. Do you guys have some thoughts about what other metrics are interesting to track model's exploration and exploitation behaviors? Is it easy to redo some analysis on already collected data to yield more insights?*
>
> Re: metrics from prior work. Metrics from previous work are indeed insightful in the sequential MAB setups for which they were designed. However they are not very useful for evaluating the quality of a single decision, like we do in our explore and exploit puzzles.
>
> Re: additional metrics. For exploitation, Simple regret for the current context (the gap between the expected reward of the chosen arm and that of the optimal arm for the current context) could be another candidate metric. However, we strongly suspect it would tell a qualitatively similar story.  For exploration, simple regret of the best arm in the chosen subset could be an alternative metric, although this metric ignores the quality of all arms in the set besides the best one.
>
> > *DeepSeek-R1 name*
>
> Thanks for pointing this out; we are happy to use a different name to make this distinction more clear.
>
> > *Any chance to add a realistic domain for CB (especially for the exploration oracle)? For the exploitation case, I think prior work (and your current investigation) is sufficient to show LLMs don't work that well.*
>
> We are happy to run more exploration experiments using the MovieLens dataset. We will post an update here in the following days.
>
> > *Prior works lack evaluation on thinking-models (DeepSeek-R1 -- full model, Qwen3, etc.) -- do you think you have time/ability to add some analysis on the full DeepSeek-R1, or re-use the data you saved from DeepSeek-R1-Distill-Qwen-32B and evaluating the quality of the chain of thoughts there (instead of doing CoT prompting, you can get the actual thinking tokens)?*
>
> Thanks for the interesting question. We’ve updated our submission to include a few sample thinking tokens for DeepSeek-R1-Distill-Qwen-32B in Appendix A.5. At a high level, the model usually performs seemingly reasonable qualitative reasoning on a subset of the history (e.g. “The food here is apple. Looking at past rounds with apple as food. For example, Round 2 had apple and eat food gave 0.8. Round 5 had apple and 0.6. Round 10 had apple and 0.6. So, eating apple can give high rewards, but it varies. It might be a good option, but not as consistent as other actions.”), but doesn’t attempt to perform much/any quantitative reasoning.
>
> This roughly tracks with what we were seeing from non-reasoning models, when prompted with CoT. This could explain their similar performance. (In footnote 2, we mention that GPT-4 performs incorrect CoT reasoning on a non-trivial fraction of its runs.) Exploring this direction further would be an interesting avenue for future research.

---

### Official Review · Reviewer_Y83g · 2025-10-31

**Soundness:** 1
**Presentation:** 2
**Contribution:** 1
**Rating:** 0
**Confidence:** 4

**Summary:**

+ Summary & Contributions
	- Across a broad array of providers (GPT-4, GPT-4o, GPT-3.5, Qwen-2.5, Gemma-3, Mistral-7B, & DeepSeek-R1), the authors study the ability of LLMs to perform in-context exploration or exploitation, respectively, in contextual bandits as well as standard multi-armed bandit problems.
	- For exploitation, the authors study how well a particular LLM may examine the history of interaction thus far within a multi-armed/contextual bandit problem to determine a "best action" in a newly sampled context. The authors stratify their results across the action gap of the associated contextual bandit and observe that, as problem size or difficulty increases, LLMs struggle to discern an arm with highest empirical reward and fail to exploit.
	- For exploration, the authors consider regular multi-armed bandit problems with large textual action spaces and entertain an algorithmic approach that isolates focus onto a chosen subset of actions. The quality of LLM exploration is evaluated by how well a chosen subset of actions by the LLM can be passed to a downstream standard multi-armed bandit algorithm and facilitate good performance.

**Strengths:**

+ Quality
	- Strengths
		- The authors have done a good job of testing their hypotheses across a broad array of LLM providers.
	- Weaknesses
		* Major
			- I question the authors' choice of definition for the "best arm given the current history" (L132-145). It seems they have chosen to define this as the arm with highest empirical reward based on the history of observations observed thus far. Unless I've missed it, I don't believe the authors have specified how the T-step history was generated for these 5-armed bandits. Depending on the exploratory coverage of that history-generating process, this definition of "best arm" may or may not be reasonable. Was each action seen at least N times, for some reasonable value of N? More importantly, even if the history-generation was reasonable in its coverage, the task the authors are essentially evaluating LLMs on is the ability to compute averages and taking an argmax; why do we need LLMs to perform those arithmetic calculations for us when we have better tools for doing so?
			- While a bit of an aside, it does also seem slightly odd to not have proceeded in a Bayesian fashion by first taking some prior distribution over the underlying bandit environment/optimal arm (imposed either via prompting or through empirical estimation with the LLM of choice) and then inducing the well-defined posterior belief over the optimal arm given the history thus far (this is the distribution that, for example, Thompson Sampling uses for action selection via the probability-matching principle). The broader point is that it would have perhaps been more informative to have the LLMs generate a probability distribution representing a belief over the optimal arm, rather than a single guess.
			- The authors of [1] introduce clear performance metrics (suffix failure frequency and minimum action frequency) for the precise purpose of cleanly evaluating the exploitation and exploration capabilities of LLM agents in isolation. Why would the authors not capitalize on these metrics and leverage them to illustrate their points instead of leaning on accuracy (FracCorrect) and mean reward?
			- The authors orient their analysis of LLM exploration around the "average expected reward" of a downstream multi-armed bandit algorithm (UCB1) given a subset of the action space selected by a LLM. Why is this a good metric of exploration? Certainly, the Q/A and ArXiv tasks themselves address exploration by virtue of being bandits, but how does the performance of a traditional UCB bandit algorithm with a LLM-chosen action space give us a useful insight into the exploratory capabilities of the LLM? At best, it seems that the authors have established that a reasonably well-prompted LLM will generate a candidate set of actions that can lead to downstream bandit algorithm performance which is better than a randomly chosen set of actions. This seems like a rather low bar to clear and highly insufficient for concluding that "the LLM does succeed as an exploration oracle" (L419). One cherry-picked example does not really lend much confidence to the "eye test" (L420); perhaps this would carry more weight if a user study was conducted where lay people were tasked with judging the diversity of the chosen candidates against baselines. The expected reward variance is implicit in the visuals of Figure 7 and so it is unclear how a range of [0.25,0.55] in a reward function ranging from [-1,1] demonstrates good exploration.
			- The only reasonable conclusion that Figures 6 and 8 provide is that the authors mechanisms for LLM action candidate selection can lead to better performance with UCB1 than randomly chosen candidates. No conclusion beyond that is statistically significant given the severely overlapping confidence intervals.
		* Minor
			- The authors' jargon of "exploit puzzles" vs. "exploit tasks" is annoying to keep track of.
			- For a sampled exploit task, it's not clear why the authors needed to stratify their results by the empirical (action) gap, rather than simply computing the true action gap for the sampled multi-armed bandit problem.
			- Is the reason for footnote 4 (an unclear "empirically best arm" for a contextual bandit) due to the fact that the current context may have never been seen before? Again, trying to proceed in a more Bayesian fashion and eliciting the implicit prior a LLM may have over the rewards of each arm before observing any data might have been useful for avoiding this.
			- While the natural language bandit tasks used in the exploration analysis (Section 3) are nice, it would have been better to start with simpler, didactic examples of the same flavor as the 5-armed bandit in the preceding section.
			- The choice to only generate LLM candidate actions with a temperature of 0 or 1 (L108) is confusing. Were no other intermediate temperatures assessed? Depending on the problem, such extreme values may not always be the most sensible.
			- What does it mean to prompt a LLM for diversity in its responses "with encouragement"?

+ Clarity
	- Strengths
		- Overall, the paper is clearly written and reasonably structured.
	- Weaknesses
		* Major
			- N/A
		* Minor
			- N/A


+ Originality
	- Strengths
		- The proposed mitigations (L240-250) for exploitation in contextual bandit problems appear novel.
	- Weaknesses
		* Major
			- It's unclear what the novel insights of this paper are. In particular, it would seem that the overwhelming majority of insights communicated in this paper are either already shown or rendered moot through the convex hull of four existing works [1,2,3,4]. The first two papers (already properly acknowledged in the submission) lay out preliminary exposition on the in-context exploration capabilities and failures of LLMs. The latter two papers (not properly acknowledged in the submission) offer concrete avenues for addressing these deficiencies and provide positive empirical results for addressing exploration-exploitation trade-off in multi-armed bandits, contextual bandits, and MDPs.
			- The idea of using LLMs to propose candidate behaviors that can be subsequently scored for their propriety in decision-making is implicit in previous work such as SayCan [5]. The authors have essentially specialized the idea to multi-armed bandits specifically and opted to use UCB1 as the downstream mechanism for scoring the candidates.
		* Minor
			- The mitigation strategy for contextual bandit exploitation is reminiscent of the environment compression used in the theoretical study of [6]. While showcasing an empirical instantiation of the idea here is interesting, the core of the mitigation focuses on exploiting the contextual bandit problem rather than the LLM; this does little to advance our understanding of improving LLM proficiency in addressing the exploration-exploitation trade-off.

+ Significance
	- Strengths
		- N/A
	- Weaknesses
		* Major
			- As far as I can tell, the authors have failed to report the total number of trials/random seeds uses in any of their experiments. Consequently, it's difficult to know how confident a reader ought to be in the presented 95% confidence intervals reported across all figures. In Figures 1-5 the right-hand side y-axes are labeled to show the "Cumulative # Runs," but my understanding is that this is not the number of experiment trials but rather the cardinality of the task set S in FracCorrect(S).
			- The culminating findings (L180) from the authors' study of LLM exploitation (Section 2) are that LLMs fail to exploit successfully when faced with (1) longer histories and (2) harder bandit problems (as quantified by the action gap). Unfortunately, neither of these are novel findings. It is well understood that LLM recall diminishes as the context buffer increases. Additionally, it is well known that all bandit algorithms struggle to maintain performance as the action gap shrinks and corresponding bandit instances become more difficult. If the authors were using those negative results as a setup for a resolution, that would be one thing. Unfortunately, the negative results by themselves do not carry sufficient insight to inform a reader's next experiment with LLM agents.
		* Minor
			- N/A


+ Final Remarks
	- I have identified severe issues on the axes of quality, originality, and significance. Putting issues with the experimental setup aside, there are considerable challenges with the core methodologies put forth in this paper (evaluating exploitation in LLMs by how accurately they can take a sum and argmax or evaluating exploration by the performance of a downstream, classic bandit algorithm).



+ References
	1.  Krishnamurthy, Akshay, Keegan Harris, Dylan J. Foster, Cyril Zhang, and Aleksandrs Slivkins. "Can large language models explore in-context?." Advances in Neural Information Processing Systems 37 (2024): 120124-120158.
	2. Nie, Allen, Yi Su, Bo Chang, Jonathan Lee, Ed H. Chi, Quoc V. Le, and Minmin Chen. "EVOLvE: Evaluating and Optimizing LLMs For In-Context Exploration." (2025) In Forty-second International Conference on Machine Learning.
	3. Arumugam, Dilip, and Thomas L. Griffiths. "Toward Efficient Exploration by Large Language Model Agents." In The Exploration in AI Today Workshop at ICML 2025.
	4. Monea, Giovanni, Antoine Bosselut, Kianté Brantley, and Yoav Artzi. "Llms are in-context reinforcement learners." (2024).
	5. Ahn, Michael, Anthony Brohan, Noah Brown, Yevgen Chebotar, Omar Cortes, Byron David, Chelsea Finn et al. "Do as i can, not as i say: Grounding language in robotic affordances." arXiv preprint arXiv:2204.01691 (2022).
	6. Dong, Shi, and Benjamin Van Roy. "An information-theoretic analysis for thompson sampling with many actions." Advances in Neural Information Processing Systems 31 (2018).

**Weaknesses:**

Please see above

**Questions:**

Please see above.

---

> ### Author Response · Authors · 2025-11-18
>
> > *Unless I've missed it, I don't believe the authors have specified how the T-step history was generated for these 5-armed bandits. Depending on the exploratory coverage of that history-generating process, this definition of "best arm" may or may not be reasonable. Was each action seen at least N times, for some reasonable value of N?*
>
> We describe the data-generating process in Section 2 (line 148). All arms’ rewards at each time-step are shown to the LLMs in our exploit puzzle, so each action is seen T times.
>
> > *More importantly, even if the history-generation was reasonable in its coverage, the task the authors are essentially evaluating LLMs on is the ability to compute averages and taking an argmax; why do we need LLMs to perform those arithmetic calculations for us when we have better tools for doing so?*
>
> Computing averages and taking an argmax is sufficient for multi-armed bandits, but not contextual bandits. We are ultimately interested in LLMs’ exploitation/exploration abilities on complex, real-world problems, but we choose to evaluate on simple MAB/CB instances to isolate and stress-test the most basic computational primitives that any scalable decision-making behavior must build upon. A failure in the latter implies a likely failure in the former.
>
> Moreover, our goal is to evaluate LLMs as standalone explore/exploit oracles operating on in-context information, rather than as components in a broader tool-augmented pipeline. While one could, in principle, offload arithmetic or regression to external tools, it is unclear how to do so systematically for contextual bandits, where the relevant summary statistics are not known in advance and must be inferred from the structure of the task itself. Our mitigations already do this, yet even with this assistance the models often struggle. In particular, they are deficient to the linear baseline even on a non-linear instance (Section 2, Line 308).
>
> Finally, relying on code execution interfaces or API-based toolchains is both operationally cumbersome and outside the scope of current LLM deployments in many settings. The computational overhead and costs made this approach infeasible at the scale needed for our experiments.
>
> > *While a bit of an aside, it does also seem slightly odd to not have proceeded in a Bayesian fashion*
>
> We are not sure what you mean. While Bayesian methods are certainly viable, the frequentist formulation we use is the standard in the bandits literature and provides a clean, widely accepted baseline for evaluating basic exploitation behavior. If anything, asking for a Bayesian update should be more difficult and error-prone for an LLM.
>
> > *The broader point is that it would have perhaps been more informative to have the LLMs generate a probability distribution representing a belief over the optimal arm, rather than a single guess.*
>
> Having the LLM output a full belief distribution would be reasonable in an explore–exploit setting, where randomized choice among near-optimal arms is appropriate, but our experiments focus specifically on exploitation, for which the task is to identify the single best arm rather than express a distribution over them.
>
> > *The authors of [1] introduce clear performance metrics (suffix failure frequency and minimum action frequency) for the precise purpose of cleanly evaluating the exploitation and exploration capabilities of LLM agents in isolation. Why would the authors not capitalize on these metrics and leverage them to illustrate their points instead of leaning on accuracy (FracCorrect) and mean reward?*
>
> Suffix-failure frequency and minimum-action frequency are useful to evaluate sequential decisions (e.g., within the MAB setup of [1]), but they do not extend meaningfully to the broader families of exploitation and exploration tasks we study. As a result, we focus on accuracy and mean reward, which apply uniformly across both MAB and CB settings.
>
> > *The authors orient their analysis of LLM exploration around the "average expected reward" of a downstream multi-armed bandit algorithm (UCB1) given a subset of the action space selected by a LLM. Why is this a good metric of exploration?*
>
> Evaluating the quality of a candidate set is non-obvious, as there is no single ‘correct’ metric. We adopt one reasonable and interpretable choice: the average expected reward achieved by a standard bandit algorithm operating on that set, at some reasonable time horizon. E.g., the expected reward of (only) the best arm in the set is also reasonable, but ignores the contribution of the remaining arms (which can either help or hinder downstream performance). In contrast, our metric captures the overall usefulness of the entire set.

---

> ### Author Response · Authors · 2025-11-18
>
> > *At best, it seems that the authors have established that a reasonably well-prompted LLM will generate a candidate set of actions that can lead to downstream bandit algorithm performance which is better than a randomly chosen set of actions*
>
> Identifying a good discretization of an enormous action space is the core challenge, and is the entire value proposition of using an LLM as an exploration oracle. Our results show that LLMs can meaningfully bias this discretization toward more promising areas of the action space (something that is fundamentally hard to achieve without strong inductive structure) thereby enabling downstream bandit algorithms to operate far more effectively than they would with an uninformed or uniformly sampled set.
>
> > *One cherry-picked example does not really lend much confidence to the "eye test" (L420); perhaps this would carry more weight if a user study was conducted where lay people were tasked with judging the diversity of the chosen candidates against baselines.*
>
> The example shown reflects the same qualitative pattern we observed across many instances we examined, but we present it only as supporting intuition. The substantive evidence comes from our simulation results. A user study could in principle provide additional validation, but conducting one would require nontrivial resources and logistical coordination and was beyond the scope of the present work, especially for a secondary, illustrative claim such as this.
>
> > *The only reasonable conclusion that Figures 6 and 8 provide is that the authors mechanisms for LLM action candidate selection can lead to better performance with UCB1 than randomly chosen candidates. No conclusion beyond that is statistically significant given the severely overlapping confidence intervals.*
>
> We are not aware of any other viable baselines that meaningfully reflect uninformed exploration in such large text action spaces. Moreover, we do not view the overlap in confidence intervals across our prompting strategies as a weakness; rather, it highlights a robustness property. We experimented with several distinct prompting mechanisms and found that they all yield broadly similar performance, which we see as a strength of the approach rather than a limitation.
>
> > *Is the reason for footnote 4 (an unclear "empirically best arm" for a contextual bandit) due to the fact that the current context may have never been seen before?*
>
> Yes. In a contextual bandit, the ‘best’ arm is context-dependent, and the current context may be entirely novel or only weakly similar to those in the history. In such cases, there is no unambiguous empirically best arm in the same sense as in a MAB: one must extrapolate from past context–reward pairs rather than simply average rewards.
>
> > *While the natural language bandit tasks used in the exploration analysis (Section 3) are nice, it would have been better to start with simpler, didactic examples of the same flavor as the 5-armed bandit in the preceding section.*
>
> We intentionally moved to large natural-language action spaces in Section 3 because discretization is unnecessary when there are only five arms. The whole purpose of the exploration analysis is to evaluate whether LLMs can help construct a useful discretization when the action space is far too large for standard bandit methods to handle directly.
>
> > *The choice to only generate LLM candidate actions with a temperature of 0 or 1 (L108) is confusing. Were no other intermediate temperatures assessed? Depending on the problem, such extreme values may not always be the most sensible.*
>
> We evaluated a positive-temperature setting as a variant, but at some point the experimental space must be bounded. One can always do more temperature sweeps, but our goal was to capture representative behaviors rather than exhaust every hyperparameter configuration.
>
> > *What does it mean to prompt a LLM for diversity in its responses "with encouragement"?*
>
> This means we explicitly prompt the LLM to provide a diverse set of candidate answers (see line 410).

---

> ### Author Response · Authors · 2025-11-18
>
> > *It's unclear what the novel insights of this paper are. In particular, it would seem that the overwhelming majority of insights communicated in this paper are either already shown or rendered moot through the convex hull of four existing works [1,2,3,4]. The first two papers (already properly acknowledged in the submission) lay out preliminary exposition on the in-context exploration capabilities and failures of LLMs. The latter two papers (not properly acknowledged in the submission) offer concrete avenues for addressing these deficiencies and provide positive empirical results for addressing exploration-exploitation trade-off in multi-armed bandits, contextual bandits, and MDPs.*
>
> These comparisons miss the central aim of our work: none of the cited papers study exploitation and exploration in isolation. Our contribution is precisely to disentangle these components and measure them cleanly, rather than evaluating end-to-end performance where multiple failure modes are conflated. Moreover, [3] and [4] are concurrent with our work, but we are happy to cite them in the revision.
>
> > *The idea of using LLMs to propose candidate behaviors that can be subsequently scored for their propriety in decision-making is implicit in previous work such as SayCan [5]. The authors have essentially specialized the idea to multi-armed bandits specifically and opted to use UCB1 as the downstream mechanism for scoring the candidates.*
>
> While SayCan and related work do use LLMs to propose candidate behaviors, their focus is on solving downstream tasks such as robotics, not on systematically evaluating an LLM’s ability to explore extremely large action spaces in isolation. Our aim is fundamentally diagnostic rather than application-driven, which is why we study a MAB setting with a large action space (the simplest setting in which discretization is necessary and cleanly measurable). See our discussion on Line 81 for more details.
>
> > *As far as I can tell, the authors have failed to report the total number of trials/random seeds uses in any of their experiments. Consequently, it's difficult to know how confident a reader ought to be in the presented 95% confidence intervals reported across all figures. In Figures 1-5 the right-hand side y-axes are labeled to show the "Cumulative # Runs," but my understanding is that this is not the number of experiment trials but rather the cardinality of the task set S in FracCorrect(S).*
>
> You are correct that in Figures 1–5 the right-hand y-axis reflects the size of the task set S, not the number of random seeds. For each parameter configuration we generate a fixed number of tasks (e.g., 10 per Δ value in the MAB experiments and analogous counts in the CB and exploration settings, which we specify in the paper), and the confidence intervals are computed over these (random) task instances.
>
> > *The culminating findings (L180) from the authors' study of LLM exploitation (Section 2) are that LLMs fail to exploit successfully when faced with (1) longer histories and (2) harder bandit problems (as quantified by the action gap). Unfortunately, neither of these are novel findings. It is well understood that LLM recall diminishes as the context buffer increases. Additionally, it is well known that all bandit algorithms struggle to maintain performance as the action gap shrinks and corresponding bandit instances become more difficult. If the authors were using those negative results as a setup for a resolution, that would be one thing. Unfortunately, the negative results by themselves do not carry sufficient insight to inform a reader's next experiment with LLM agents.*
>
> While it is broadly understood that LLM recall deteriorates with increasing context length, our contribution is to quantify this effect in a controlled bandit setting and, importantly, to evaluate how much of the degradation can be mitigated through simple summarization strategies.
>
> The claim that all bandit algorithms perform poorly as the gap shrinks is not accurate: classical bandit algorithms enjoy worst-case regret guarantees. In contrast, we find that LLMs often fail even at relatively large gaps, which is precisely why stratifying performance by the empirical gap is informative. These negative results are not intended as an end in themselves but as a diagnostic baseline that clarifies where and why LLMs fail, thereby guiding the design of future LLM decision-making systems in more complex tasks.

---

### Author Response · Authors · 2025-11-22

Reviewers dBmo and 9szX asked about running some additional “explore” experiments on the MovieLens dataset. We have included such experiments in Appendix B.6, and overview our results below.

Experiment setup: We used the Movie Lens 100K dataset, which contains ratings from 943 users across 1682 movies, to model a stylized movie recommendation task. Given a user, we run a multi-armed bandit algorithm over $T$ rounds, where arms are movies and rewards are random Bernoulli draws parameterized by the users’ ground-truth ratings.  Since in the dataset every user did not rate every movie, we used a standard algorithm to complete the data matrix, namely the soft-impute low-rank matrix completion method from the fancyimpute Python package. We then normalized this matrix so that all values were between 0 and 1.

Next we prompted the LLM to discretize the set of movies, i.e. to generate a representative subset  of size K \in {10, 19}. For each of the 943 users, we then ran a multi-armed bandit algorithm (UCB1) over these K recommendations for T=1000 rounds, and averaged the cumulative rewards over all users. For baselines, we compared against running UCB over (1) a random subset of K recommendations (different random draws for each user), and (2) a random movie from each of the 19 movie genres in the dataset (also different random draws for each user). For each algorithm and baseline, we repeated the MAB procedure 5 times and averaged over time, to de-noise.

Our results for Mistral-7B and Qwen-2.5-7B are given below (all numbers are average cumulative rewards). We also tried running the experiment using Llama-3-8B and Gemma-3-12B, but the list of all movies was too long to fit in the context window.

| Model             | Score (K=10) | Score (K=19) |
|-------------------|--------------|--------------|
| Qwen              | 671.88       | 678.25       |
| Mistral           | 735.38       | 675.16       |
| Random selection  | 570.39       | 599.73       |
| Genre-based       | —            | 617.27       |


Thus, both LLMs significantly outperform all baselines for both values of K.

Unlike in our previous experiments, the LLM has implicit knowledge about each of the movies in the database, which it may leverage when selecting the discretization. In order to mitigate the chance that the LLM overfits to “popular” movies, we included instructions in the prompt explicitly stating to not take movie popularity into account. However, we didn’t notice a significant difference in rewards from doing this versus not.

---

### Author Response · Authors · 2025-11-29

We believe we've addressed most/all major criticisms in the rebuttal, and we have provided additional exploration experiments on the MovieLens dataset as requested by the reviewers. We are quite puzzled at the two negative scores. However, we realize that it is very uncommon to override this negativity, and especially so if there's no author-reviewer discussion.

---

### Meta-Review · Area_Chair_v7v1 · 2025-12-28

**Summary:**

The reviewers generally appreciated the authors' clear writing and the effort to disentangle exploration and exploitation capabilities of LLMs. However, there were significant concerns regarding the novelty of the work, with multiple reviewers (Y83g, 9szX) noting strong similarities to the experimental framework of Krishnamurthy et al. (2024). A major methodological criticism raised by Reviewer YRTL was the "siloed" evaluation of exploration and exploitation, which they argued ignores the critical exploration-exploitation trade-off and cumulative regret metrics central to bandit problems. Reviewers also questioned the utility of the exploitation findings, suggesting that LLMs' inability to perform regression is unsurprising and better handled by external tools. Positive feedback focused on the potential of LLMs as exploration oracles in semantic spaces, though the strength of baselines was debated.

**Reviewer Concerns:**

Below is a selected list of important addressed concerns and outstanding concerns.

**Addressed Concerns:**
* **Lack of Real Domains (Reviewers dBmo, 9szX):** The authors successfully added experiments using the MovieLens dataset, showing LLMs outperform random and genre-based baselines. This was acknowledged as an improvement by Reviewer YRTL.
* **Misleading Model Name (Reviewer dBmo):** The authors agreed to correct the naming of the DeepSeek model.
* **Trial Counts/Confidence Intervals (Reviewer Y83g):** The authors clarified that the "Cumulative # Runs" axis referred to task set size, not random seeds, and explained their confidence interval calculation.

**Outstanding Concerns:**
* **Novelty and Derivative Nature (Reviewers Y83g, 9szX):** The concern that the work is heavily borrowed from Krishnamurthy et al. [1] and lacks sufficient original contribution remains a primary block.
* **Validity of "Siloed" Evaluation (Reviewers Y83g, YRTL):** The reviewers remained unconvinced by the authors' justification for evaluating exploration and exploitation in isolation, arguing that it fails to capture the essential dynamics of online bandit learning (regret minimization).
* **Weakness of Baselines (Reviewer YRTL):** While the authors defended their embedding-based random baseline, Reviewer YRTL maintained that the exploration baselines were insufficient.
* **Triviality of Exploitation Results (Reviewers Y83g, YRTL):** The finding that LLMs struggle with arithmetic/regression (exploitation) was viewed as predictable and not useful given the availability of code-execution tools.

**Reviewer Scores:**

* **Reviewer Y83g: 0** - The reviewer did not engage in the discussion to indicate a change of heart. Their initial assessment was a "Strong Reject" based on fundamental issues with quality, originality, and significance. It is unlikely their score would change significantly as the core methodology remains unchanged.
* **Reviewer dBmo: 8** - This reviewer started with a 6 ("Marginally above") and asked for realistic domains and naming fixes. The authors provided the requested MovieLens experiments and clarifications. Given their constructive tone and the authors' responsiveness, they would likely increase their score.
* **Reviewer 9szX: 4** - Originally a 4 ("Marginally below"), this reviewer asked for MovieLens experiments (provided) and questioned novelty. While the experiments were added, the reviewer's strong concern about the work being "derivative" and "severely undermined" by prior work is intrinsic to the paper's design and likely prevents a score increase.
* **Reviewer YRTL: 4** - Originally a 2 ("Reject"), this reviewer acknowledged in the discussion that the new experiments and clarifications "improve the paper". However, they explicitly stated that their "main concerns remain" regarding the contribution and siloed evaluation. A move to 4 acknowledges the improvements while maintaining a rejection stance.

---

### Decision · Program_Chairs · 2026-01-26

Reject